

**Vegetation-climate feedbacks modulate rainfall patterns in Africa under**
**future climate change**
M. Wu[1], G. Schurgers[2], M. Rummukainen[1,3], B. Smith[1], P. Samuelsson[4], C. Jansson[4], J.
Siltberg[1], W. May[3]
[1]{Department of Physical Geography and Ecosystem Science, Lund University, Sölvegatan 12, SE-
223 62, Lund, Sweden}
[2]{Department of Geosciences and Natural Resource Management, University of Copenhagen,
Øster Voldgade 10, DK-1350 Copenhagen, Denmark}
[3]{Centre for Environmental and Climate Research, Lund University, Sölvegatan 37, SE-223 62 Lund,
Sweden}
[4]{Rossby Centre, Swedish Meteorological and Hydrological Institute, SE-601 76, Norrköping,
Sweden}
Correspondence to: M. Wu (minchao.wu@nateko.lu.se)



## *Abstract*

Africa has been undergoing significant changes in climate patterns and vegetation in recent decades, and continued changes may be expected over this century. Vegetation cover and composition impose important influences on the regional climate in Africa. Climate change-driven changes in regional vegetation patterns may feed back to climate via shifts in surface energy balance, hydrological cycle and resultant effects on surface pressure patterns and larger-scale atmospheric circulation. We used a regional Earth system model incorporating interactive vegetation-atmosphere coupling to investigate the potential role of vegetation-mediated biophysical feedbacks on climate dynamics in Africa in an RCP8.5-based future climate scenario. The model was applied at high resolution (0.44 x 0.44 degrees) for the CORDEX-Africa domain with boundary conditions from the CanESM2 GCM. We found that changes in vegetation patterns associated with a $CO_2$ and climate-driven increase in net primary productivity, particularly over sub-tropical savannah areas, not only imposed important local effect on the regional climate by altering surface energy fluxes, but also resulted in remote effects over central Africa by modulating the land-ocean temperature contrast, Atlantic Walker circulation and moisture inflow feeding the central African tropical rainforest region with precipitation. The vegetation-mediated feedbacks were in general negative with respect to temperature, dampening the warming trend simulated in the absence of feedbacks, and positive with respect to precipitation, enhancing rainfall reduction over rainforest areas. Our results highlight the importance of vegetation-atmosphere interactions in climate projections for tropical and sub-tropical Africa.



**Keywords**: RCA-GUESS, Vegetation feedback, Precipitation, Walker Circulation, land-
ocean contrast


























## *1. Introduction*


The Sahel greening and Congo rainforest browning observed since the 1980s suggest that
Africa has been undergoing significant vegetation changes in structure and composition
during the recent decades (Zhou et al., 2014;Eklundh and Olsson, 2003;Olsson et al.,
2005;Jamali et al., 2014). In addition to influences from anthropogenic activity (e.g. changes
in land use), vegetation shifts in the region have been linked to changes in recorded climatic
conditions including the trend and interannual variability of precipitation (Herrmann et al.,
2005;Olsson et al., 2005;Zhou et al., 2014;Hickler et al., 2005), which in turn have been
related to decadal-scale changes in regional circulation (Camberlin et al., 2001;Giannini et al.,

88 2003).

Shifts in vegetation composition, cover and seasonality (phenology) can in turn impose
significant feedbacks on the physical climate system by altering surface-atmosphere energy
exchange and hydrological cycling (biophysical feedbacks) as well as greenhouse gas
concentrations and aerosol loads in the atmosphere (biogeochemical feedbacks). The type
and coverage of vegetation are important to surface albedo, roughness length and
evapotranspiration, affecting surface energy fluxes that in turn control lower boundary layer
thermodynamics (Bonan, 2008;Brovkin et al., 2006;Eltahir, 1996). Biophysical feedbacks
operate locally and may also generate teleconnections via heat and moisture advection,
leading to altered circulation patterns (e.g. Avissar and Werth, 2005;Nogherotto et al., 2013).
There is an increasing awareness of the significance of biophysical vegetation-climate
interaction for Africa. Vegetation changes in the Sahel can alter local decadal-scale
precipitation variability through changes in local energy and water fluxes and even through
changes in atmospheric circulation (Charney, 1975;Xue and Shukla, 1996;Wang and Eltahir,


2000), while deforestation in the Congo basin increases surface albedo and weakens local
moisture convection, resulting in decreased precipitation (Eltahir, 1996;Xue and Shukla,
1993;Bell et al., 2015;Nogherotto et al., 2013).
Anthropogenic climate change may lead to profound structural and compositional changes
in the natural vegetation over Africa, especially for savannah areas where seasonal
fluctuations in water availability and climate-mediated disturbance regimes (fires and
grazing) serve to facilitate coexistence of trees and grasses in a fine competitive balance
(Moncrieff et al., 2014;Sankaran et al., 2005;Doherty et al., 2010;Ahlström et al., 2015).
Changed vegetation patterns may be expected to modulate the regional climate
development. However, high-resolution studies of future vegetation-atmosphere coupling
have not been performed earlier for Africa with a comprehensive approach.
We employ a regional Earth system model (ESM) that couples the physical component of a
regional climate model (RCM) with a detailed, individual-based dynamic vegetation model
(DVM). This tool enables dynamic representation of biophysical interactions between the
vegetated land surface and the atmosphere and their effects on the evolution of climate
and land surface biophysical properties to be analysed in an explicit way. We perform
simulations under Representative Concentration Pathway (RCP) 8.5 future radiative forcing
(Moss et al., 2010) with and without vegetation feedbacks enabled, and investigate the
potential coupled evolution of climate and vegetation patterns for the CORDEX-Africa
domain over the 21$^{st}$ century. Our focus is especially on the central African rainforest areas
and the surrounding savannah vegetation belt.

## 2. Data and Method

### 2.1   Model description



RCA-GUESS (Smith et al., 2011) is a regional ESM based on the Rossby Centre regional
climate model RCA4 (Kjellström et al., 2005;Samuelsson et al., 2011) coupled with
vegetation dynamics from the LPJ-GUESS DVM to account for land-atmosphere biophysical
coupling (Smith et al., 2001;Smith et al., 2014).
The RCA4-based physical component of the model incorporates advanced regional surface
heterogeneity, such as complex topography and multi-level presentations for forests and
lakes, which are significant in controlling the development of weather events from the local-
to meso-scale (Samuelsson et al., 2011).  RCA4 has been successfully applied in a range of
climate studies worldwide (e.g., Sörensson and Menéndez, 2011;Kjellström et al.,
2011;Döscher et al., 2010). The land surface scheme (LSS, Samuelsson et al., 2006) adopts a
tile approach and characterizes land surface with open land and forest tiles with separated
energy balance. The open land tile is divided into fractions for vegetation (herbaceous
vegetation) and bare soil. The forest tile is vertically divided into three sub-levels (canopy,
forest floor and soil). Snow can exist in open land and/or forest tile as fractional cover.
Surface properties such as surface temperature, humidity and turbulent heat fluxes (latent
and sensible heat fluxes) for different tiles in a grid box are weighted together to provide
grid-averaged surface boundary conditions. A detailed description is given by Samuelsson et
al. (2006).
The vegetation dynamics component of RCA-GUESS employs a plant individual and patch-
based representation of the vegetated landscape, optimized for studies at regional and
global scale. Heterogeneities of vegetation structure and their effects on ecosystem
functions such as carbon and water vapour exchange with the atmosphere are represented
dynamically, affected by allometric growth of age-size classes of woody plant individuals,



along with a grass understorey, and their interactions in competition for limited light and
soil resources. Plant functional types (PFTs) encapsulate the differential functional
responses of potentially-occurring species in terms of growth form, bioclimatic distribution,
phenology, physiology and life-history characteristics. Multiple patches in each vegetated
tile account for the effects of stochastic disturbances, establishment and mortality on local
stand history (Smith et al., 2001). This explicit, dynamic representation of vertical structure
and landscape heterogeneity of vegetation has been shown to result in realistic simulated
vegetation dynamics in numerous studies using the offline LPJ-GUESS model (Smith et al.,
2001;Hickler et al., 2012;Smith et al., 2014;Wårlind et al., 2014;Wu et al., 2015;Weber et al.,
2009). Biophysical feedbacks have previously been studied in applications of RCA-GUESS to
Europe and the Arctic (Wramneby et al., 2010;Smith et al., 2011;Zhang et al., 2014). A
general description for the coupling between the vegetation dynamics component LPJ-
GUESS and the physical component RCA is provided in the Appendix. A more detailed
description is given by Smith et al. (2011).
## *2.2    Model setup, experiments and analysis approach*
The simulations were applied over the African domain of the Coordinated Regional Climate
Downscaling Experiment (CORDEX-Africa, Giorgi et al., 2009;Jones et al., 2011) on a
horizontal grid with a resolution of 0.44° × 0.44°. The period studied was 1961 to 2100.
Forcing (atmospheric fields and sea-surface temperature as lateral and lower boundary
conditions) followed the historical and RCP8.5 simulations with the CanESM2 general
circulation model (GCM) (Arora et al., 2011) in the Coupled Model Intercomparison Project
Phase 5 (CMIP5, Taylor et al., 2012).



The vegetation sub-model LPJ-GUESS was set up with eight global PFTs which represent the
major groups of natural vegetation across Africa, including the tropical and warm-temperate
forests and $C_3$ and $C_4$ grass. The characteristics for the PFTs are based on the study by
Morales et al. (2007). They are summarised in Table A1.
PFTs in the forest tile were simulated with 30 replicate patches. Average values of state
variables across the replicate patches were used to determine biophysical parameters, i.e.
forest fraction and leaf area index (LAI) for trees versus grasses, provided as forcing to the
physical part of the model. For the open land tile with herbaceous species, $C_3$ and $C_4$ grass
were simulated deterministically and aggregated to characterise open land vegetation. Fire
disturbance in response to climate and simulated fuel load (Thonicke et al., 2001) was
included.
Following the approach of Wramneby et al. (2010) and Smith et al. (2011), RCA-GUESS was
initialized with a spin-up in two stages to achieve a quasi-steady state representative for
mid-1900's conditions. After the spin-up, the model was run in coupled mode from 1961
onwards, with simulated dynamic meteorological conditions from the physical sub-model
affecting vegetation phenology and structural dynamics, and biophysical land surface
properties being adjusted to reflect the changes in vegetation, thereby affecting the physical
climate dynamics. For comparison, a recent past experiment (RP, Table 1) with the same
vegetation spin-up but then driven by the boundary condition from ECMWF re-analysis
(ERA-Interim) project (Berrisford et al., 2009), was conducted for the period 1979-2011.
A simulation protocol was designed for inferring biophysical feedbacks of vegetation
changes to the evolving 21st century climate. Three simulations were performed to
investigate vegetation-climate feedbacks under future climate change (Table 1). The first



simulation included the vegetation feedback (FB). It was run for 1961-2100 in coupled mode,
allowing the effects of climate and atmospheric $CO_2$ concentration (the latter was taken
directly from the RCP 8.5 data set) on vegetation state to feed back to the evolving climate.
The second simulation was run without the vegetation feedback (NFB). It started with the
state of FB simulation at 1991 and used a prescribed climatology of daily vegetation for
1961-1990 from the coupled simulation, but without allowing the simulated changes in
vegetation in LPJ-GUESS to feed back to the simulated climate in RCA. To investigate the
importance of the $CO_2$ physiological effects on vegetation changes under future climate
change, we performed a third simulation (FB_CC), which was similar to FB, but starting from
the state of FB simulation of 1991 and using historical atmospheric $CO_2$ forcing until 2005
and constant afterward for the vegetation sub-model.
In the analysis, we focus on the future period 2081-2100 and compare this with the present-
day (1991-2010). The climate change signal is inferred from the difference between the
future mean and the present-day mean in the NFB run. Vegetation feedbacks are calculated
as the difference between the future means of the FB and NFB runs. These approaches were
applied for the entire study, unless specified otherwise.

### *2.3    Methods to evaluate model performance*

Simulated near-surface atmospheric temperature over open land, precipitation, and
vegetation variable leaf area index (LAI) were compared against observations within the
common available time period 1997-2010. Temperature and precipitation were compared
with gridded observations from the CRU TS3.21 (Harris et al., 2014) dataset, focusing on the
annual mean and seasonality. For precipitation we also employed the GPCP (Huffman et al.,
2001, version 1.2 of One-Degree Daily product for 1996/10-2011/6) which uses satellite data



to upscale rain gauge measurements and has been extensively used for African precipitation
studies (e.g., Nikulin et al., 2012). For the LAI evaluation we used the GIMMS-AVHRR and
MODIS-based LAI3g product (Zhu et al., 2013) which has been previously applied for the
evaluation of vegetation dynamics in ESMs (e.g., Anav et al., 2013).
To identify biases propagating from the model physics per se and from the GCM-derived
boundary forcing data, we compared the reanalysis-driven RP simulation against
observation and against the GCM-driven (CanESM2) FB simulation for the same period.

## 3. Results

### 3.1    Model evaluation

To evaluate the model's performance for the present day, the simulated annual mean and
seasonality of 2-meter air temperature, precipitation and LAI are compared against the
observations (Fig. 1 and Fig. 2). The simulated annual mean temperature (Fig. 1a1) is
generally higher in northern-hemisphere Africa than in the south. The model generally
shows a cold bias in the order of 1°C for northern and southern savannah (Fig. 1a2),
dominated by the boreal summer (Fig. 2a1,2a3). Warm biases occur over northern Africa up
to around 3°C, as well as in central Africa (around 2°C) where the warm bias originates
mainly from summer (Fig. 2a2).
The simulated precipitation is largest over western and central Africa up to 1600 mm year$^{-1}$
within the simulated rainbelt between 25°N and 25°S, where the Atlantic moisture inflow
(monsoon and equatorial westerlies) plays an important role (Fig. 1b1). Comparison with
CRU reveals a considerable dry bias (-600 mm year$^{-1}$) for the central African rainforest area
and a wet bias (+300 mm year$^{-1}$) for the northern savannah. A comparison of the FB



(CanESM2-driven) and the RP (ERA-Interim-driven) simulations (Fig. 1b3) indicates that
apart from the uncertainty from RCM, the bias in simulated precipitation can be partly
explained by the uncertainty from the boundary conditions. The simulated patterns and
magnitude of precipitation for this area are similar to a previous study using an earlier
version of RCA, RCA3.5, without dynamic vegetation (Nikulin et al., 2012). RCA3.5 was able
to capture the main features of the seasonal mean rainfall distribution and its annual cycle,
and the model biases were of similar magnitude to the differences between observational
datasets (Nikulin et al., 2012).
The biases in simulated precipitation for the savannah regions and the central African
rainforest area mirror the temperature biases: warm biases coincide with dry biases in
central Africa, and cold biases coincide with wet biases in savannah regions. Apart from the
model uncertainty, observation uncertainty may contribute to the biases, which can be seen
when compared with GPCP (Fig. 2b1, 2b2): For the northern savannah, CRU tends to present
lower precipitation than GPCP and the modelled throughout the year. For the central
African rainforest area, precipitation from CRU is considerably higher in the mid-year dry
season, but lower for the rest of the year with much more moderate monthly precipitation
variability than in GPCP and the modelled. In general, the simulated precipitation presents a
better consistency with GPCP than with CRU, although it is difficult to evaluate the
uncertainties between these two observational datasets.
The simulated seasonality of LAI generally reflects the simulated seasonality of precipitation.
A systematic overestimation is apparent for savannahs, and an underestimation for the
central Africa rainforest area. These biases in LAI predominantly reflect the corresponding
biases in precipitation (Fig. 2 b1-b3 and 2c1-c3). A stronger LAI bias in the savannah can be



explained by the presence of grasses, which are more sensitive to precipitation changes in
the model compared to trees.
With present-day forcing, the simulated climate and vegetation patterns and phenology are
generally consistent with observed patterns. Some of the biases in the simulated climate are
common to most RCMs (Nikulin et al., 2012) and they are apparent for some sub-regions
and seasons in our model. However, we consider the performance adequate to capture the
main details of the African climatology, which provides sufficient confidence for the
subsequent analysis of regional vegetation-climate interactions under future climate change.
### 3.2    *Future climate and vegetation change*
In the simulation without vegetation feedbacks (NFB) under the RCP8.5 scenario and
CanESM2 boundary conditions, most of the African continent is simulated to be 4-6°C
warmer by the end of the 21$^{st}$ century (Fig. 3a). The subtropics exhibit a slightly stronger
warming than the tropics, and land warming is slightly larger than warming of the
surrounding ocean surface (note that the SSTs were prescribed from the GCM). These
changes are fairly similar throughout the year, except in Northern Africa and the Sahara,
where the temperature increase is particularly pronounced in the local dry season (Fig. A1e-
h). Precipitation is projected to increase in most parts of the African monsoon area, western
equatorial coastal area and the eastern African horn (Fig. A2e-h). A slight decrease is
projected in the Congo basin and for the southern part of the continent (Fig. 3c). For areas
with a precipitation increase, the increase is mainly confined to the local wet season. The
precipitation decrease over central and southern Africa is apparent throughout the year (Fig.
A2e-h).



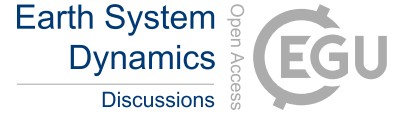

Vegetation feedbacks (FB) modify significantly the pattern and magnitude of simulated
climate change. The effects are largest in low-latitude areas where the surface temperature
increase is generally dampened (negative feedback), most notably in savannah areas and to
a lesser extent in the equatorial rainforest area (Fig. 3b). The precipitation decrease is
enhanced (positive feedback), most notably over rainforest (Fig. 3d).
With the effects of climate change and $CO_2$ fertilization, future vegetation growth depicts an
enhancement not only of vegetation productivity in general, but also of tree cover in
subtropical savannah areas (Fig. 4a), displacing grasses and resulting in an increase in tree
LAI of 0.5-2.4 during the growing season (Fig. 4b). This increase in tree cover reflects a
general rise in vegetation productivity driven by rising atmospheric $CO_2$ concentration on
photosynthesis and water-use efficiency (Long, 1991;Hickler et al., 2008;Keenan et al.,
2013). Results from the FB_CC experiment in which $CO_2$ fertilisation is disabled reveal that
changes in climate drivers alone are simulated to have minor or opposing effects on tree
productivity and LAI due to reduced water availability (Fig. A3), and that the changes above
hence originate primarily from $CO_2$ fertilization.
Temperature feedbacks tend to be strong in the newly afforested areas (Fig. 3b, Fig. 4a). The
cooling effects from vegetation feedbacks are strong (approximately -2°C) throughout the
year, with the most pronounced cooling occurring in the local dry season (Fig. A1i-l), when
the newly established forest (with larger root depth than grass) transpires water that is
taken up from the deeper soil layer. Transpiration from present-day grass is constrained by
the low moisture levels in the top soil layer. As a result, the evaporative cooling effect
becomes stronger when forest replaces open land. In the central African rainforest area with
increase of LAI by about 0.5-1, vegetation feedbacks on temperature are much smaller in the



rainy season, but cause considerable cooling effect for the dry season.
Vegetation feedbacks on precipitation are also pronounced. For the Southern hemisphere
savannah area, a slight increase in precipitation (approximately 10%, Fig. 3d) was simulated,
which is caused by strengthened convective activity (which coincides with enhanced
radiation and latent heat fluxes) in the rainy season (DFJ, Fig. A2). This can be considered as
a local effect from forest expansion. However, changes in precipitation are not restricted
only to the areas where forests expand (Fig. 3d, Fig. 4a), which is suggestive of remote
effects for tropical precipitation. This is further investigated in the sections below.

### 3.3    *Vegetation feedback effects on circulation and precipitation*

Vegetation feedbacks on temperature in our simulations operate mainly via an increased
surface area for evaporation and a stronger coupling to the atmosphere as tree cover, root
depth and LAI increase relative to grasses, most notably in savannah areas, resulting in a
shift of the evaporative fraction (ratio of latent heat flux to turbulent heat fluxes) and an
increase in surface roughness length. Overall, the turbulent heat fluxes increase, which
tends to cool the surface and the lower atmosphere. Similar behaviour was seen in southern
Europe a feedback study with RCA-GUESS (Wramneby et al., 2010).
The variability of precipitation over Africa is greatly influenced by the moisture advection
from the ocean to land. Studies have noted on the influence of Atlantic Walker circulation on
central African precipitation, as well as the role of the west African monsoon (WAM) for the
precipitation for western Africa (e.g. Nicholson and Grist, 2003;Nicholson and Dezfuli,
2013;Dezfuli and Nicholson, 2013). These circulation systems are associated with thermal
contrasts between ocean and land, creating a pressure contrast that tends to promote the





movement of moist surface air from the Atlantic over land. We examined the land-ocean
thermal contrast (∇T) and geopotential contrast (∇ϕ) between the equatorial Atlantic and
the near-coast African continent for three pressure levels between 850 hPa and 975 hPa, to
investigate the circulation in the lower troposphere. We found that changes in ∇T and ∇ϕ are
highly inter-annually anti-correlated for the rainy seasons MAM and SON (r=-0.82 and -0.64,
respectively, Fig. 5; Fig. A4). The sensitivity of ∇ϕ to ∇T, depicted as the slope in Fig. 5, is
generally maintained in the future, with a slight decrease in the sensitivity for DJF and a
slight increase for MAM.
Under the NFB future simulation, ocean-land contrast becomes larger (the absolute value of
∇T increases by about 0.5-1°C, Table A2) as land temperature increases more than the ocean
surface temperature (Fig. A1), due to differential changes in features of the surface and
lower atmosphere, such as changes in land-ocean contrasts in boundary layer lapse rate
(Joshi et al., 2008) and changes in Bowen ratio over land (Sutton et al., 2007). As a result,
except for SON, ∇ϕ is generally simulated to increase in the course of the simulation (Fig.
A4), with the largest shift occurring in MAM (11.96 $m^2 s^{-2}$ by the end of 21$^{st}$ century, Table
A2). For SON, ∇T increases but ∇ϕ does not, suggesting that the trend of ∇ϕ under climate
change is associated with the GCM-derived boundary conditions, despite the strong regional
coupling with ∇T in terms of variability (Fig. A4).
In contrast, the increase in the ∇T is dampened considerably when incorporating interactive
vegetation. The resulting reduction in ∇T offsets ∇ϕ uniformly and statistically significantly
for all seasons, generally counteracting the climate change effect on ∇ϕ (Fig. 5, Table A2).
***3.4    Effects on Walker circulation and low-latitude precipitation***



The low-level equatorial westerlies are important to the central African rainfall. They are
associated with the lower branch of the Walker cell located near the western equatorial
coast of Africa, and they transfer moisture from the adjacent Atlantic to the eastern
equatorial coast and the Congo basin (e.g. Schefuß et al., 2005;Nicholson and Grist, 2003).
These westerlies occur from March to October, being best developed in JJA.  They shift
northward with the excursion of the Inter Tropical Convergence Zone (ITCZ) and under the
strong influence of the South Atlantic high pressure cell (Nicholson and Grist, 2003). This
pattern is simulated by RCA-GUESS for the present-day climate (Fig. 6). Via this circulation
system, moisture can reach far over the African landmass at around 28°E, upwell and
integrate into the mid-level African Easterly Jet (AEJ) (Camberlin et al., 2001;Nicholson and
Grist, 2003). RCA-GUESS reproduces this pattern with a realistic magnitude (Fig. 6, Fig. 7, Fig.
8, Fig. 9) when compared with previous studies based on reanalysis data (Camberlin et al.,
2001;Nicholson and Grist, 2003).
In the NFB future simulation, equatorial westerlies are strengthened throughout the year
both over ocean (Fig. 6) and over land (Fig. 7). Changes in wind speed ($\Delta u$) can be explained
by changes in the low-level pressure contrast between land and ocean (sect. 3.3), where
strengthened $\nabla\phi$ leads to enhanced u, especially for MAM when the zonal pressure contrast
prevails (Table A2). Atmospheric specific humidity in the lower troposphere near the equator
also increases by around 10%-20% for MAM and SON, extending from the ocean to inland
along the equator (Fig. 8cd; Fig. 9cd). Meanwhile, changes in future rainfall are apparent
along the equator, with increases over the equatorial coastal or inland areas (Fig. A2),
concurrent with stronger moisture inflow to land in the low-level troposphere (Fig. 8cd; Fig.
9cd).



Vegetation feedbacks are simulated to weaken the climate change enhancement of the
Walker circulation, resulting in a weakening of the equatorial westerlies and counteracting
the effects of climate change alone (Fig. 6i-l and Fig. 7i-l; Fig. 8ef and Fig. 9ef). These changes
correspond well to changes in low-level ocean-land geopotential contrast $\Delta\nabla\phi$ with the
biggest impact for MAM and SON (Table A2). The weakened Walker circulation is also
represented as suppressed vertical uplifting motions over central Africa (Fig. 8f and Fig. 9f).
Atmospheric specific humidity at 850 hPa is reduced by approximately 7% due to vegetation
feedbacks which are comparable to the contribution of climate change (Fig. 8ef vs. Fig. 8cd;
Fig. 9ef vs. Fig. 9cd).
Analysis of the moisture flux convergence also confirms the impacts of a weakened Walker
circulation (Fig. 10) on the hydrological cycle caused by vegetation feedback. Moisture fluxes
for most parts of the African continent diverge toward the ocean near the equatorial
regions. This divergence is similar for both MAM and SON but the effect is slightly stronger
for SON, which also corresponds to reduced humidity for these areas (Fig. 8e-f; Fig. 9e-f).
The changes in precipitation show a distinct spatial and temporal pattern with changes in
the rainbelt area (defined as 2mm day$^{-1}$ contour with 10-days smoothing, Fig. 11). Under
future conditions, the rainbelt, which follows the ITCZ excursion, shifts around 3° northward
during JAS (Fig. 11a). As a result, rainfall intensity increases from May to October, with the
most pronounced increase by more than 30% relative to present-day levels of around 2 mm
day$^{-1}$ on the margins of the rainbelt.  The rainy season becomes longer for Sahel (+9 days) as
well as for central Africa (+1 day). The location of the rainbelt for the rest of the year
remains unchanged, but there is a pronounced increase in rainfall intensity for southern
African rainy season (about 10%) and a decrease (about -10%) for the central African rainy



seasons.
On top of the non-feedback climate change effect, vegetation feedbacks tend to cause a
slight contraction of the rainbelt around the equator, and they impose a primarily
counteractive effect on rainfall intensity compared to the climate change alone simulation
(NFB). For central Africa, the considerable decrease in rainfall intensity in the dry season
leads to a slight equatorward shrinking of the rainbelt (approximately 2°) and a shorter rainy
season (on average 10 days, represented as a 4-day postponed onset and a 6-day earlier
end). For southern Africa, strengthened convective precipitation results in a longer rainy
season by on average 6 days. There is no pronounced effect for the Sahel regions except for
some sparse changes over time and in some areas. To investigate the effects on ITCZ
location, we analysed the position of the intertropical front (ITF) with a meridional wind
criterion (Sultan and Janicot, 2003) by examining the location of maximum vertical uplifting
wind speed at 850 hPa over Sahel in July and over southern Africa in January. However, we
did not find pronounced effects for ITF (not shown) suggesting that changes in the rainbelt
location for central Africa are mainly caused by changes in precipitation intensity rather
than by changes in meridional circulation.

### 4. Discussion

#### 4.1    Related tenets of Regional Earth System Modelling

Previous studies on land-climate interactions for Africa were carried out either over some
African sub-region (e.g. Wang and Alo, 2012;Yu et al., 2015), or at a relatively coarse
resolution within the implementation of GCMs (e.g. Kucharski et al., 2013), or without
considering feedback effects from natural vegetation dynamics and only investigating
anthropogenic impacts such as deforestation or afforestation (Lawrence and Vandecar,



2015). In this study, we investigated the coupled dynamics of climate and vegetation
patterns over Africa under a future climate change scenario, applying a regional-scale ESM.
The development of regional-scale coupling, including vegetation dynamics in a coherent
way, enables the quantification of vegetation-change-induced feedbacks in climate
simulations (Rummukainen, 2010;Smith et al., 2011;Giorgi, 1995;Zhang et al.,
2014;Wramneby et al., 2010). In this way, it is able to isolate the regional biophysical
feedbacks, which are usually not easy to disentangle in a global application in which the
effects of changes in carbon-cycle and large-scale circulation tend to combine with the
biophysical effects.
In comparison with global ESMs, the added value from the regional ESMs lies in the
enhanced resolution obtained in a regional setup as presented in this study, allowing for a
more detailed representation of small-scale surface features such as topography, land use,
vegetation change, and consequently possible related feedbacks, and also enhancing the
model's ability to capture climatic variability and extreme climatic events (Rummukainen,
2010). In addition, improvements in the representation of local processes (e.g. those that
determine surface temperature) may result in improved larger scale features (e.g. sea level
pressure) (Feser, 2006;Diffenbaugh et al., 2005). As seen also in a previous evaluation study
for the atmosphere-only version of RCA for Europe, a reduced bias in surface air
temperature results in a better representation of interannual variability of mean sea level
pressure and circulation patterns, and improves the simulation of precipitation (Kjellström
et al., 2005).
**_4.2     Vegetation dynamics over Africa for present and future_**



Vegetation dynamics are critically important in modulating the evolution of the 21st century
climate in our study. Land use and grazing (Lindeskog et al., 2013;Bondeau et al.,
2007;Sankaran et al., 2005), which were not included in our study, represent additional
drivers of land surface changes. The historical vegetation state is also relevant for future
simulations, due to legacy effects lasting decades or even centuries (Wang et al.,
2004;Moncrieff et al., 2014). Apart from the biases in climate (model) forcing, biases in
simulated vegetation may come from the absence of consideration of these aspects and
result in over- or under-estimation of the vegetation state. Nevertheless, the vegetation-
feedback effects associated with the vegetation sub-model are still likely to be captured
here, as vegetation dynamics in terms of forest cover changes and interannual and inter-
seasonal variability of vegetation productivity are more important than the absolute
vegetation state when considering vegetation feedbacks. Previous studies in the offline
vegetation model LPJ-GUESS suggests that the vegetation dynamics for savannah and
tropical forest vegetation are robust (Weber et al., 2009;Ahlström et al., 2015), providing
additional confidence for the examination of the vegetation-climate interaction in our study.
Under future climate change, the vegetation response to environmental changes will differ.
As revealed by previous experimental (Kgope et al., 2010) and modelling (Sitch et al.,
2008;Moncrieff et al., 2014) studies, vegetation may be expected to become less sensitive
to climate conditions when the atmospheric $CO_2$ concentration increases. This is because
$CO_2$ fertilization of photosynthesis and enhanced water use efficiency linked to a reduction
of stomatal conductance, which causes a shift towards higher woody-plant dominance,
resulting in densification of tree and shrub cover relative to grasses in savannahs, or
replacement of savannah with forest. Shrub encroachment and woody thickening has been





observed in water-limited areas including Sahel in recent decades, coinciding with rising $CO_2$
concentrations (e.g. Liu et al., 2015). In our results, the simulated vegetation dynamics are
consistent with these trends, presenting a similar trajectory of vegetation changes (not
shown), and a similar vegetation pattern (Fig. A3) as in previous modelling studies (e.g.,
Sitch et al., 2008;Moncrieff et al., 2014).
### *4.3    Vegetation feedbacks and land-ocean temperature contrasts*
The land-ocean contrast is an important driver of continental precipitation, as it determines
the transport of moisture from ocean to land (e.g. Lambert et al., 2011;Fasullo,
2010;Giannini et al., 2005;Boer, 2011;Giannini et al., 2003). The recent change in Sahel
rainfall is a good example of linking moisture transport to land-ocean contrast, where
changes in SSTs over adjacent tropical oceans around Africa are key to the fragile balance
that defines the regional circulation system (Rowell, 2001;Giannini et al., 2003;Camberlin et
al., 2001). Land-surface feedback is found to modify the interannual to interdecadal climate
variability in this region by vegetation-induced albedo or evapotranspiration effects
(Charney, 1975;Zeng et al., 1999;Wang et al., 2004). In our study, the SSTs were prescribed
from CanESM2, therefore the altered land-ocean thermal contrast between simulations
with and without feedback originated solely from the changes in land surface temperature
induced by vegetation dynamics. Although this represents a land-forced mechanism in
contrast to an ocean-forced one in other studies (e.g. Giannini et al., 2003;Tokinaga et al.,
2012), we assume that the mechanisms are similar given the similarity in the magnitude of
simulated circulation changes. Wind speed and land-ocean temperature contrast change
approximately by 0.2 m s$^{-1}$ and 0.2°C, respectively, between FB and NFB in our study (Fig. 5
and Table A2); these are comparable to the changes simulated in other studies for the Sahel



(approximately 0.2-0.5 m s$^{-1}$ per 0.2°C (Giannini et al., 2005)) and for the Pacific Oceans
(approximately 0.3 m s$^{-1}$ per 0.3°C (Tokinaga et al., 2012)). However, the relative importance
of such changes may differ for local climate systems: the lower branch of the Walker cell
over the eastern tropical Atlantic Ocean, which we have focused on in this study, may be in
a fragile balance and is more vulnerable to changes in thermal contrasts (equatorial
westerlies slowed down by approximately 0.2 m s$^{-1}$ from less than 2 m s$^{-1}$ of the present-day
wind speed in rainy seasons, Table A2) compared to the stronger monsoonal circulation for
Sahel and the Walker cell over the equatorial Pacific Ocean (> 5 m per second wind speed in
their peak months, Young, 1999). Our results indicate that even a small disturbance of the
eastern Tropical Atlantic circulation cell may produce profound impacts (larger relative
reduction in precipitation compared with the studies by Giannini et al. (2005) *and* Tokinaga
et al. (2012)). Moreover, we assume that a study with a dynamic ocean component would
result in a similar outcome, as the ocean heat capacity is relatively large and variation in
land-ocean thermal contrast can be greatly buffered by ocean heat uptake (Lambert and
Chiang, 2007).

## 5. Conclusion and outlook

In this study we investigated the potential role of vegetation-mediated biophysical
feedbacks on climate change projections for Africa in the 21$^{st}$ century. In current savannah
regions, enhanced forest growth results in a strong evaporative cooling effect. We also
identify alterations in the large-scale circulation induced by savannah vegetation change,
resulting in remote effects and modulation of tropical rainfall patterns in Africa. Our results
emphasize the importance of accounting for vegetation-atmosphere interactions in regional
climate projections for tropical and sub-tropical Africa and stress the necessity to consider



vegetation feedbacks for more reliable estimates of regional future climate change
projections.
Future work can include detailed studies on the role of vegetation feedbacks in the regional
climate projections with respect to shorter-term dynamics such as climate variability and
extreme events, which may have crucial implications for land surface processes such as
wildfire. On the other hand, regional and global biogeochemical feedbacks on future climate
change may be triggered by regional biophysical feedbacks, which can impose important
influences to regional climatic trend, variability and seasonality in conjunction with future
greenhouse forcing. Such changes may impose important influences on the African carbon
balance, especially for the semi-arid ecosystem like savannahs whose carbon balance is
strongly associated with changes in climatic conditions (Ahlström et al., 2015). Further
examination of vegetation feedback effect in regional ESMs may have its distinct values
especially when regional processes are concerned. Future work should also include the
ongoing development of ESMs including the improvement of  the system's ability to
represent land surface properties by incorporating important land surface processes such as
changes in land use (e.g. forest clearing/grazing) and land management (e.g. irrigation), for
use in versatile land-atmospheric interaction studies.






*Appendix A: Description for the coupling between RCA and LPJ-GUESS*
In RCA-GUESS, the LSS in RCA is coupled with LPJ-GUESS, which feeds back vegetation
properties to RCA. RCA provides net downward shortwave radiation, air temperature,
precipitation to LPJ-GUESS. In return, LPJ-GUESS provides daily updated LAI and the annually
updated tile sizes (determined from the simulated maximum growing season LAI summed
across tree and herbaceous PFTs in the previous year (Smith et al., 2011)). In the forest tile
in RCA, vegetation cover in this tile is estimated as the foliar projective cover (FPC) using
Beer's law:
$$A_{tree} = 1.0 - \exp(-0.5 \cdot LAI_{tree}), \qquad (1)$$
where $LAI_{tree}$ is the aggregated LAI of woody species, simulated by LPJ-GUESS in its forest
tile in which vegetation is assumed to comprise trees and understory herbaceous vegetation.
The natural vegetated faction of the open land tile was calculated similarly:
$$A_{grass} = 1.0 - \exp(-0.5 \cdot LAI_{grass}), \qquad (2)$$
where $LAI_{grass}$ is the summed LAI of the simulated herbaceous PFTs from the herbaceous
tile of LPJ-GUESS in which only herbaceous vegetation is allowed to grow. The relative
covers of the forest and open land tiles affect surface albedo, which is a weighted average
of prescribed albedo constants for forest, open land and bare soil and controls the
absorption of surface incoming solar radiation, and therefore influences surface energy
balance and temperature.
The turbulent heat fluxes are influenced by the properties of each tile, such as surface
roughness and surface resistance, which partly depend on vegetation properties provided
by LPJ-GUESS. The vegetation surface resistance controls vegetation transpiration and bare
soil evaporation for latent heat flux calculation. It scales with LAI and varies between the



different types of vegetation and affected by the incoming photosynthetically active
radiation, soil-water stress, vapour pressure deficit, air temperature and soil temperature.
The aerodynamic resistance controls both latent heat flux and sensible heat flux and is
influenced by surface roughness length distinguished from open land and forest. The total
heat fluxes and heat transfer determine the time evolution of the surface temperature and
thus the thermodynamics in the lower boundary layer. More details about the LSS  are given
in Samuelsson et al. (2006), and the description of its coupling to the vegetation sub-model
is provided by Smith et al. (2011).
Table A1. Characteristics of the plant functional types (PFTs) used in the vegetation sub-model LPJ-GUESS.

| Characteristics | NE | BE | TrBE | TrBR | TBS | IBS | C3G | C4G |
|---|---|---|---|---|---|---|---|---|
| Leaf phenology[a] | E | E | E | D | D | D | R | R |
| Drought tolerance | low | low | low | low | low | low | very low | very low |
| Shade tolerance | high | high | high | low | high | low | low | Low |
| Optimal temperature range for photosynthesis (°C) | 10-25 | 15-35 | 25-30 | 25-30 | 15-25 | 10-25 | 10-30 | 20-45 |
| Min $T_c$ for survival (°C)[b] | - | 1.7 | 15.5 | 15.5 | -18 | - | - | 15.5 |

Notes: NE, needleleaved evergreen tree; BE, broadleaved evergreen tree; TrBE, tropical broadleaved
evergreen tree; TrBR, tropical broadleaved raingreen tree; TBS, shade-tolerant broadleaved summergreen tree;
IBS, shade-intolerant broadleaved summergreen tree; C3G, C3 grass or herb; C4G, C4 grass or herb;
[a]E, evergreen; D, deciduous; R, raingreen.
[b]$T_c$ = mean temperature (°C) of coldest month of year.






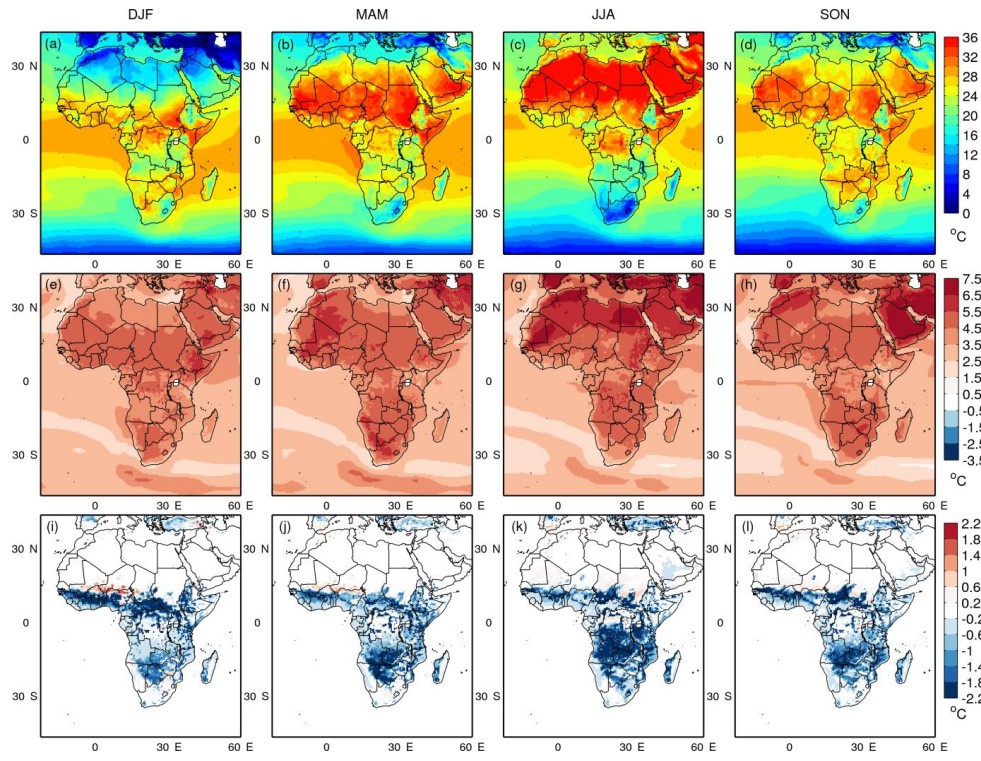

Fig. A1. Seasonal surface temperature: 1$^{st}$ panel, for present day; 2$^{nd}$ panel, changes in future in the NFB
experiment; 3$^{rd}$ panel, changes from vegetation feedback, represented as FB minus NFB for future.





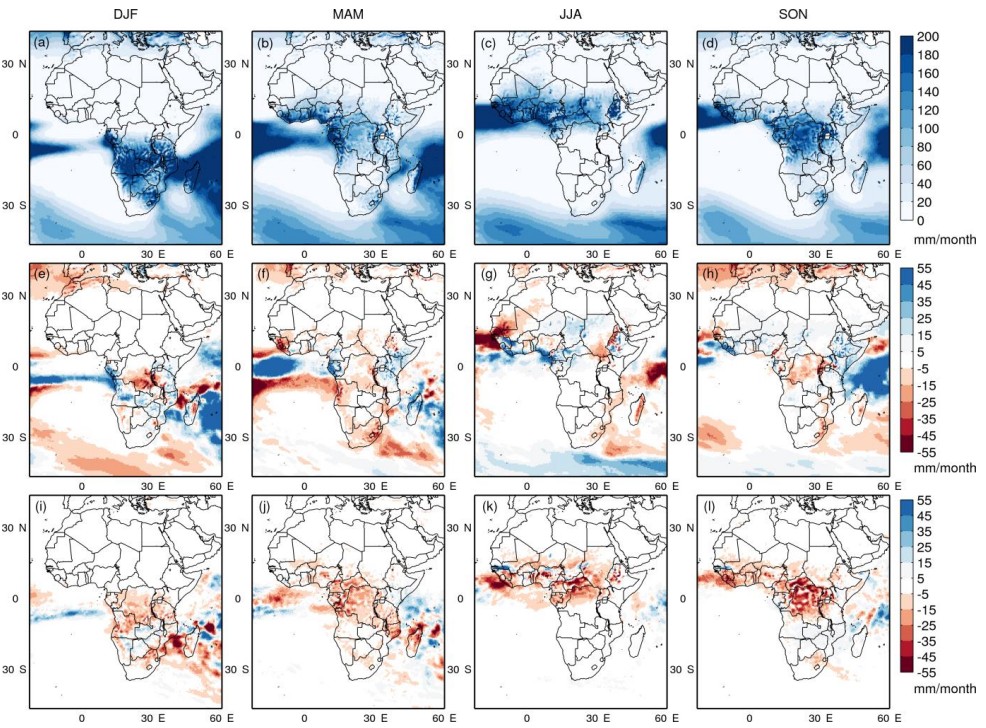


Fig. A2. Similar to Fig. A1, but for precipitation.

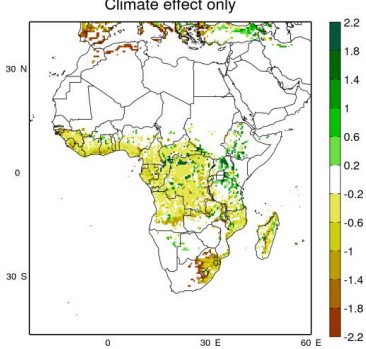


Fig. A3. Changes in forest tile LAI from the period 1991-2010 to the period 2081-2100 in FB_CC experiment.





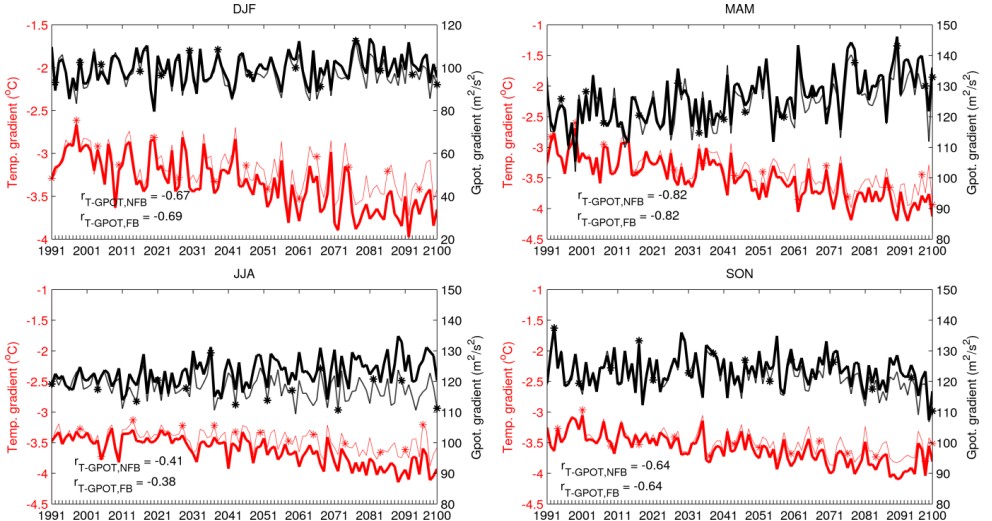


Fig. A4. Annual changes in atmospheric ocean-land temperature contrast ($\nabla T$) and geopotential contrast ($\nabla\phi$) in time series for four seasons, represented by the mean contrast at the three pressure levels 850, 925 and 975 hPa (ocean minus land) within the domain 15°N-15°S, 24°W-20°E (see the inset in the panel for JJA in Fig. 5). Correlation coefficient (r) between atmospheric temperature contrast ($\nabla T$) and geopotential contrast ($\nabla\phi$) are computed based on the de-trended annual time-series values for both FB (thick lines) and NFB (thin lines with asterisks) simulations. Changes between FB and NFB are significant at 95% confidence level for the whole time period. Note the different y-axis for DJF.

Table A2. Atmospheric temperature contrast, geopotential contrast and westerlies wind speed for the present-day state and contributions from climate change (CC subscript) and vegetation feedbacks (FB subscript), standard deviation is in parenthesis.

|  | DJF | MAM | JJA | SON |
|---|---|---|---|---|
| $\nabla T_{present-day}$ (°C) [a] | -3.06 (0.30) | -3.15 (0.34) | -3.47 (0.22) | -3.37 (0.24) |
| $\Delta\nabla T_{CC}$ (°C) [a] | -0.59[*] | -0.73[*] | -0.45[*] | -0.47[*] |
| $\Delta\nabla T_{FB}$ (°C) [a] | 0.29[*] | 0.23[*] | 0.31[*] | 0.22[*] |
| $\nabla\phi_{present-day}$ (m² s⁻²) [a] | 98.14 (5.92) | 120.86 (7.03) | 120.94 (3.83) | 124.08 (4.58) |
| $\Delta\nabla\phi_{CC}$ (m² s⁻²) [a] | 3.94 | 11.96[*] | 4.73[*] | -3.32 |
| $\Delta\nabla\phi_{FB}$ (m² s⁻²) [a] | -4.93[*] | -3.86[*] | -8.96[*] | -3.92[*] |
| $u_{zonal,present-day}$ (m s⁻¹) [b] | 0.01 (0.27) | 1.47 (0.32) | 0.87 (0.37) | 1.22 (0.31) |
| $\Delta u_{zonal,CC}$ (m s⁻¹) [b] | 0.35[*] | 0.32[*] | 0.68[*] | 0.17[*] |
| $\Delta u_{zonal,FB}$ (m s⁻¹) [b] | -0.00 | -0.21[*] | -0.28[*] | -0.16[*] |

Note: [a]: Calculations are same as Fig. 5.
[b]: $u_{zonal}$ is the averaged zonal wind speed for the pressure levels 850, 925 and 975 hPa between 3.5°N-6.5°N and 0-10°E;
The positive represents westerly and the negative represents easterly.
[*]: Changes are significant at 95% confidence level using Mann-Whitney U-test (Hollander and Wolfe, 1999).




*Acknowledgement*
This study is a contribution to the strategic research areas Modelling the Regional and
Global Earth System (MERGE) and Biodiversity and Ecosystem Services in a Changing Climate
(BECC). MCW would like to thanks Paul Miller for his helpful discussion and comments on
this work. This work was performed at the National Supercomputer Centre (NSC) in
Linköping, Sweden.

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



*Figures and Tables*

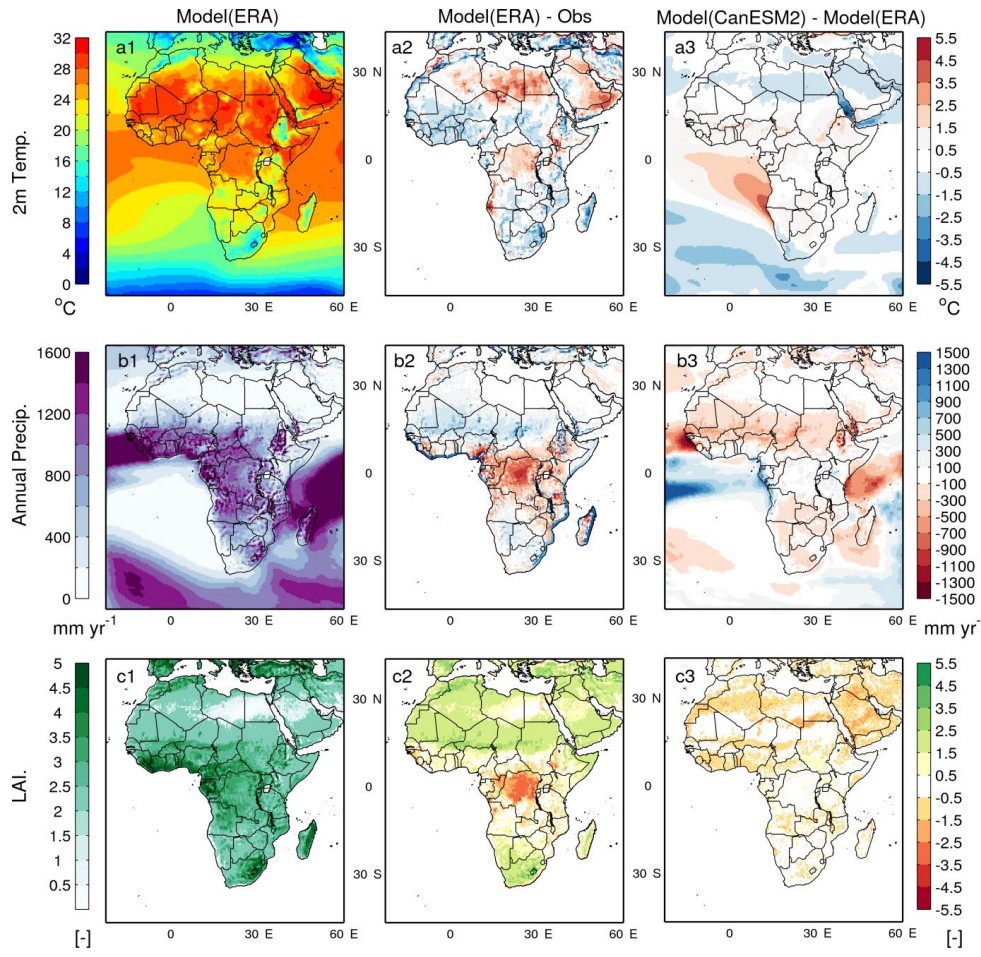


Fig. 1. Comparison between simulated and observed (a) annual mean near-surface air temperature, (b)
annual precipitation and (c) annual maximum LAI for the period 1997-2010. Variables from the RP experiment
(a1-c1) are compared with observations (a2-c2) and with those from the FB experiment (a3-c3), using RP
minus observation and FB minus RP. For the comparison with observations (a2-c2), we used CRU temperature
(a2) and precipitation (b2), as well as LAI3g (Zhu et al., 2013)(c2).



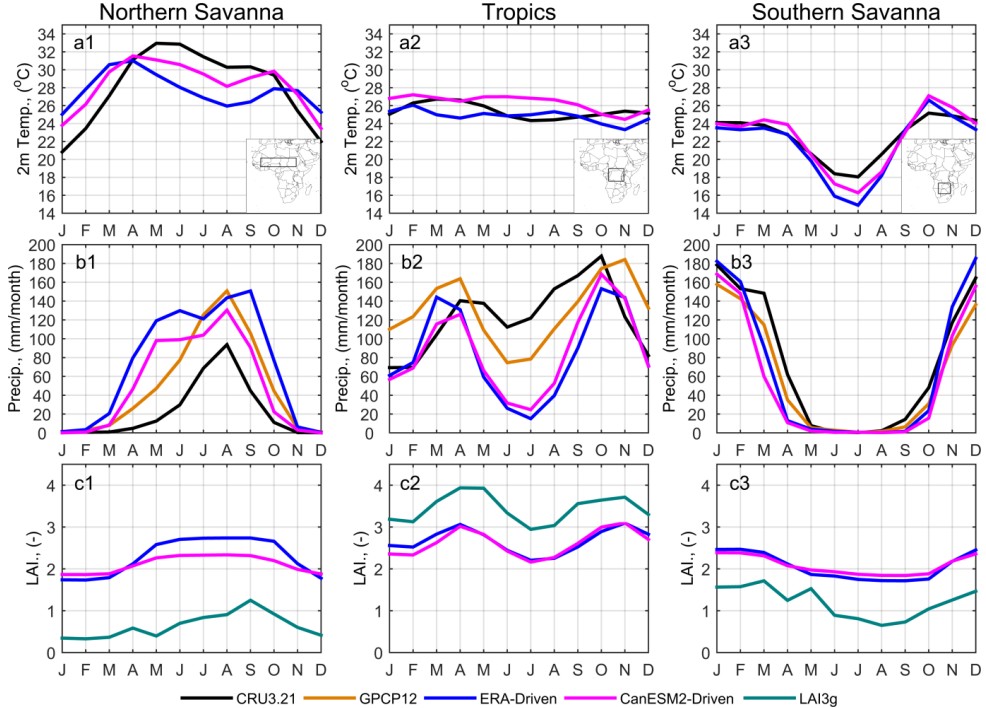


Fig. 2. Simulated seasonal cycle and observations for northern savannah (inset in a1), central Africa (inset
in a2) and southern savannah (inset in a3) for the period 1997-2010. 2m temperature (a1-a3) and precipitation
(b1-b3) are as Fig. 1. For LAI (c1-c3) monthly mean tile-weighted simulated LAI over the averaging period are
used to compare with the observation.





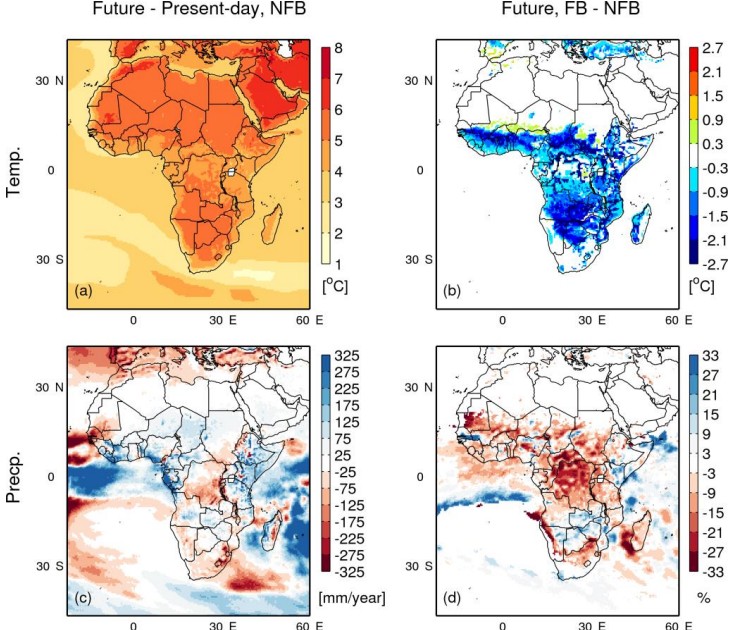

Fig. 3. Changes in surface temperature and precipitation due to climate change and vegetation feedback. The calculation of climate change signal and vegetation feedbacks, present-day and future periods are defined in Sect. 2.2. For (d), the percentage is calculated as the difference between FB and NFB (vegetation feedback) divided by the present-day level and multiplied by 100. Grid points with annual mean precipitation <20 mm year$^{-1}$ are skipped.

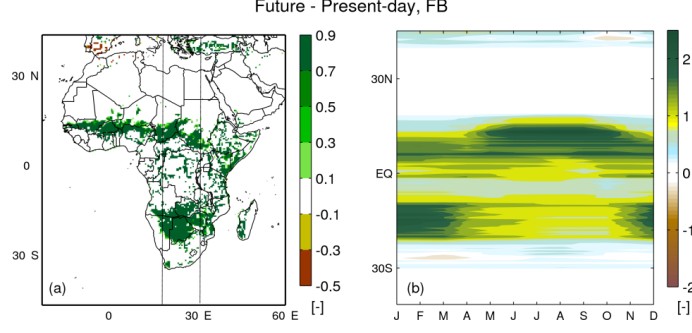

Fig. 4. (a) Change in forest fraction and (b) seasonal change in zonal mean forest LAI in the longitude band between 18°E and 30°E (lines in a), calculated as future minus present-day in FB experiment. Present-day and future periods are defined in Sect. 2.2.



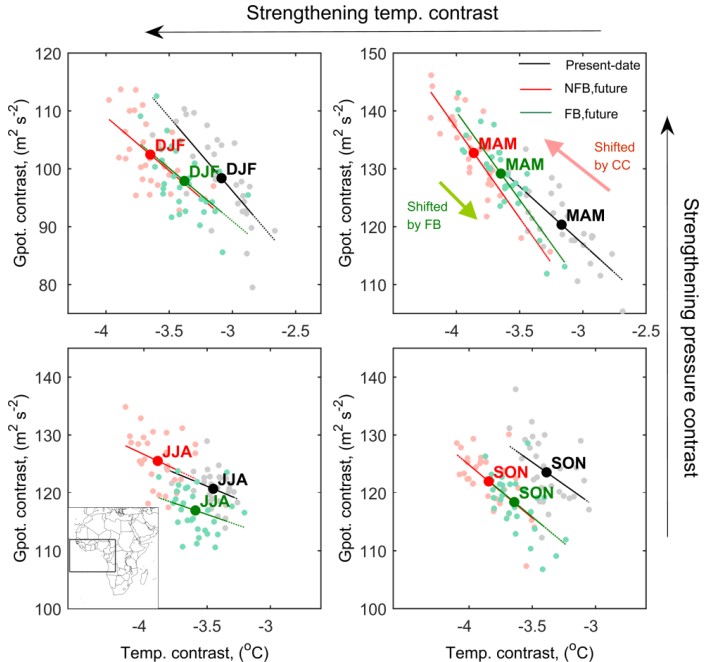

Fig. 5. Changes in atmospheric ocean-land temperature contrast ($\nabla T$) and geopotential contrast ($\nabla\phi$),
represented by the mean contrast at the three pressure levels 850, 925 and 975 hPa (ocean minus land) within
the domain 15°N-15°S, 24°W-20°E (see the inset in the panel for JJA), for the NFB and FB simulation in the
present-day and the future period (as defined in Sect. 2.2). Each scatter point represents the relation between
$\nabla\phi$ and $\nabla T$ for the correspondent season of one year, and the slopes represent its sensitivity during the
selected periods.

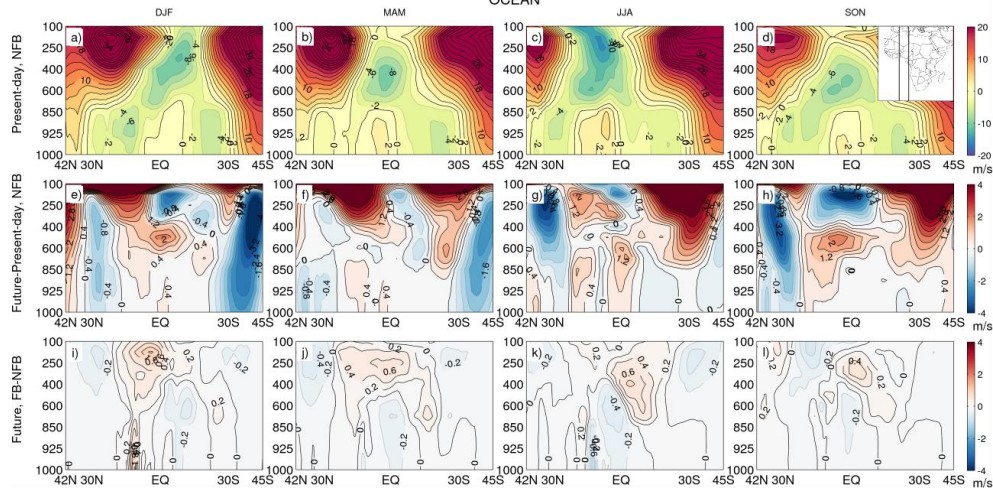

Fig. 6. Seasonal mean zonal wind speed in a cross section over adjacent Atlantic ocean (0-10°E, see the
inset in d), for present-day (1st row), changes in future (future minus present-day, 2nd row) and the
differences between FB and NFB runs in future (FB minus NFB, 3rd row). Unit is m s$^{-1}$, positive values represent



westerlies and negative values represent easterlies. Present-day and future periods are defined in Sect. 2.2
Contour intervals from top row to bottom row are 2m s⁻¹, 0.4m s⁻¹ and 0.2m s⁻¹, respectively.

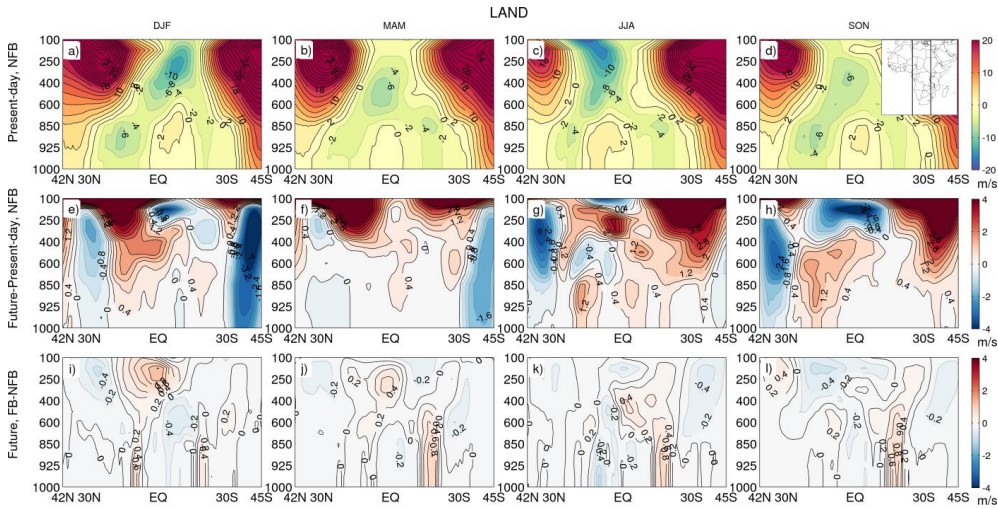


Fig. 7. As Fig. 6 but for longitudinal band over land (10°E-30°E, see the inset in d).





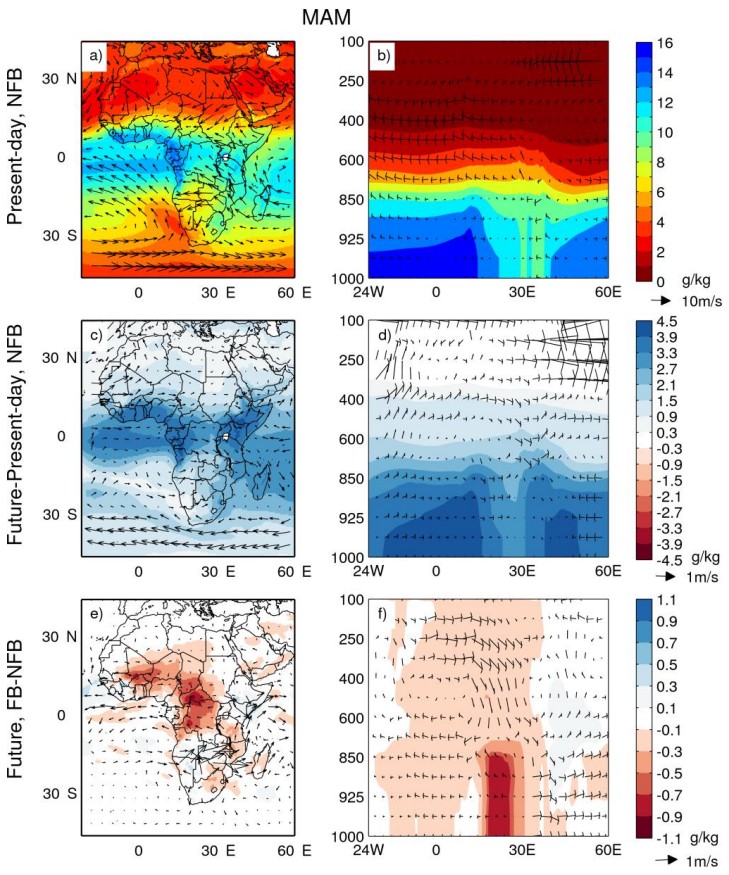


Fig. 8. Atmospheric circulation (arrows, m s$^{-1}$) and specific humidity (colour contours, g kg$^{-1}$) at 850 hPa
pressure level for MAM, displayed as (a, c, e) for the entire domain, and (b, d, f) as a cross section for a latitude
band between 2.5°S and 2.5°N, for present day (top), climate change impacts (middle) and the vegetation
feedback (bottom). Definitions for calculation period, climate change signal and vegetation feedbacks are
given in Sect. 2.2.



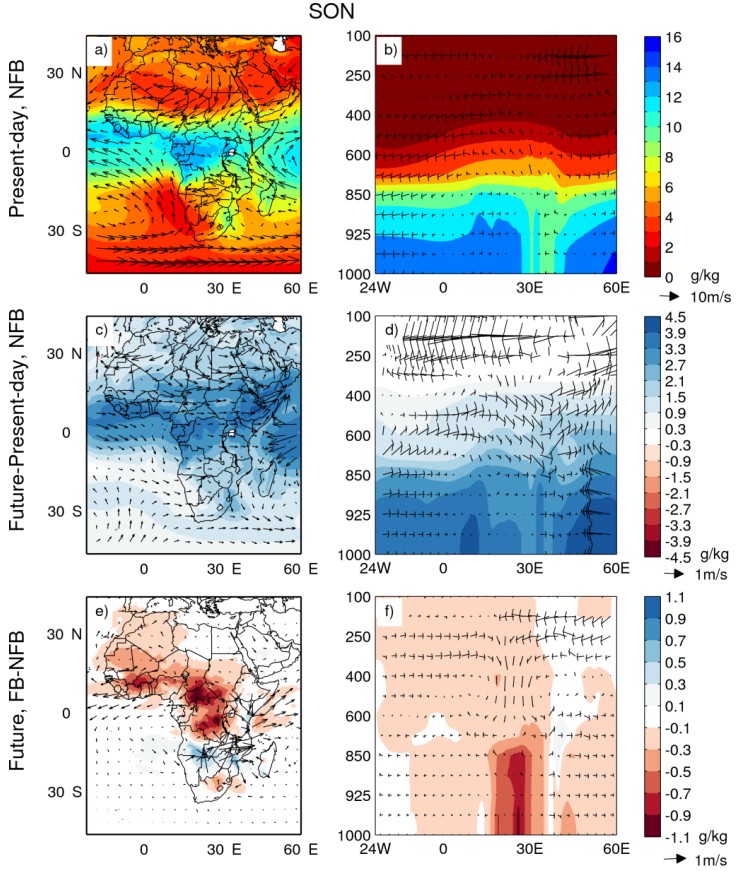


Fig. 9. As Fig. 8 but for SON.

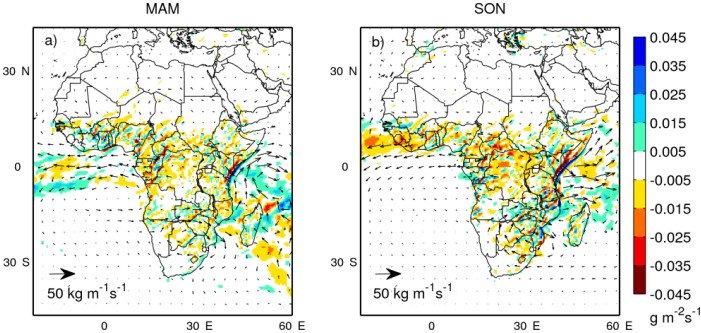


Fig. 10. Changes in vertically integrated moisture flux (arrows, kg m$^{-1}$s$^{-1}$) and moisture flux convergence
(colour contours, g m$^{-2}$s$^{-1}$) caused by vegetation feedback, averaged over the future period (as defined in Sect.
2.2) for (a) MAM and (b) SON.



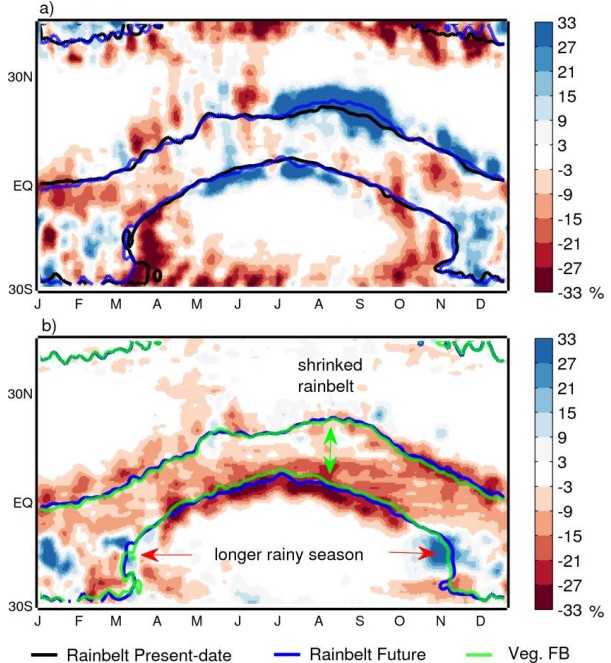


— Rainbelt Present-date  — Rainbelt Future  — Veg. FB

Fig. 11. Daily changes in precipitation averaged over the longitude band 18°E-30°E, represented as relative
changes in daily precipitation intensity (shading, %) and rainbelt location (contour) due to (a) climate change
and (b) vegetation feedback for future. The rainbelt location is defined as 2mm day$^{-1}$ contour. 10-day running
mean is applied for daily values.

Table 1. Experimental design for the investigation of the vegetation-climate feedbacks in this study.

| Runs | Vegetation Feedbacks | Radiative forcing[a] | $CO_2$ forcing[b] for vegetation sub-model | Simulated period | Boundary condition |
|---|---|---|---|---|---|
| RP | Dynamic | Historical | Historical | 1979-2011 | ERA-Interim |
| FB | Dynamic | Transient under RCP8.5 | Transient under RCP8.5 | 1961-2100 | CanESM2 |
| NFB | Prescribed vegetation simulated from 1961 to 1990 | Transient under RCP8.5 | Transient under RCP8.5 | 1991-2100 | CanESM2 |
| FB_CC | Dynamic | Transient under RCP8.5 | Historical until 2005 and constant afterward | 1991-2100 | CanESM2 |

Notes: a, using equivalent atmospheric $CO_2$ concentration; b, using actual atmospheric $CO_2$ concentration.


