# Peer review of "Vegetation-climate feedbacks modulate rainfall patterns in Africa under"

_Earth System Dynamics, 2015_

## Referee Comment (RC1) · Anonymous Referee #1 · 22 Mar 2016

This paper presents analysis of simulations with a regional climate model constrained by an earth system model (ESM) and coupled with a dynamic vegetation model (DVM). In the 21st c. simulation forced with the RCP8.5 scenario, including vegetation feedback led to drying in central Africa.

A clarification on the model setup is needed. The authors mention (lines 478-481) that "SSTs were prescribed from CanESM2, therefore the land-ocean thermal contrast . . . originated solely from the changes . . . induced by vegetation dynamics". As far as I can see, this is the last mention of the prescribed SSTs in the paper, but exactly what are these prescribed SSTs? Climatology? For what period? Given the known sensitive of the West and central African climate to SSTs, this needs to be explained carefully and the SSTs prescribed need to be evaluated. A coupled model such as the Can ESM2 is not necessarily producing correct SSTs for the observational period. Of particular

concern for the region of the analysis is the seasonal formation of the Atlantic cold tongue which, I believe, generally fails to form in coupled GCMs.

The biases in the regional model are significant (Fig. 1). The dry bias in the Congo Basin in the regional model (Fig. 1. b2), while common in models, seems extreme but it is similar to the dry bias in the ESM (Fig. 1.b3). It is important to consider how these biases influence the results, especially since one of the big results is additional drying in central Africa.

The wet bias in the Sahel in the regional model is unusual – many models fail to bring rainfall into the Sahel, as is the case for the ESM that is providing boundary conditions for the regional simulation (Fig. 1.b3). Is it relevant to the results that the regional model over-produces rainfall primarily in the spring?

I am puzzled by the large differences between the GPCP and CRU precipitation observations shown in the Sahel ("northern savanna") Fig. 2.b1. I think this is more related to the choice of averaging region than to a disparity in the observations, given the difference in the resolution of these 2 data sets. Please check this.

References to the Charney (1975) and related studies are problematic since the idea that vegetation changes (i.e., "over-grazing") caused the precipitation decline in West Africa during the 1960's and 70's has been thoroughly refuted in the mode modern literature. It's SSTs forcing, of course.

The authors note (lines 240-242) that "The simulated patterns and magnitude of precipitation for this area are similar to a previous study using an earlier version of RCA, RCA3.5, without dynamic vegetation". So doesn't that mean that dynamic vegetation is not influential, in contrast to the findings of this paper?

I would appreciate seeing an evaluation (e.g., a comparison with the ERAI reanalysis) of the circulation and specific humidity at 850 hPa wind and specific humidity from the present day, NFB simulation since the authors are pointing to changes in the circulation/moisture advection as relevant. This seems more crucial than evaluating LAI, for example.

There's not a lot of literature on the dynamics of the Walker circulation in this region and its sensitivity to SSTs (and/or land/sea contrast), but these recent papers will help:

Pokam WM, Djiotang LAT, Mkankam FK, 2012: Atmospheric water vapor transport and recycling in equatorial central Africa through NCEP/NCAR reanalysis data. Climate Dyn. 38, 1715-1729.

Pokam MW, Bain CL, Chadwick RS, Graham R, Sonwa DJ, Kamga FM, 2014: Identification of processes driving low-level westerlies in West Equatorial Africa. J. Climate 27, 4245-4262.

Cook, K. H., and E. K. Vizy, 2015: The Congo Basin Walker Circulation: Dynamics and Connections to Precipitation, Climate Dynamics, DOI 10,1007/s00382-015-2864-y.

A couple of minor points:

Please note that "Savannah" is the city in Georgia, U.S., while "savanna" is the grassland.

Figure A1 caption needs to be improved to provide more detail about what is plotted.

---

## Referee Comment (RC2) · Anonymous Referee #2 · 23 Mar 2016

This paper presents a future prediction study on climate-vegetation interactions in Africa. While the concept is not new, it does add to an emerging body of literature on interactive vegetation-climate predictions and will be of interest to many readers of ESD. The paper potentially merits publication, but quite a few major issues need to be addressed:

(1) Introduction: The flow of thought is very hard to follow. Part of the reason has to do with a rather liberal use of terminology. Probably a more strict use of the words "change" "variability" "pattern" "feedback" will help. The way it is now, many sentences are either vague or not accurate, which does not serve the readers well. Needs a better organization.

(2) Introduction: An important body of literature (e.g., Claussen 1997 climate dynamics,

[Figure]

Claussen 1998 global change biology; Zeng et al., 1999 science; Alo & Wang, 2010 climate dyamics; Yu et al., 2015 climate dynamics ) on vegetation-climate interactions is missing, although some of them are later mentioned in the Discussion section. The introduction part of a paper should be the place where the status of science is conveyed and gaps identified. Otherwise it will be misleading for readers who are new to the topic.

(3) Partly related to (2), the statement in lines 111-112 is misleading. The first several sentences in section 4.1 should be moved here to provide readers an accurate description of the status of science, and the authors need to further elaborate to explain why this study adds values to existing literature.

(4) Fig.1: The color scale is very difficult to read if one were to try to figure out the actual magnitude of the model biases. Should use more distinguishable color scales/ use stronger contrast between the colors.

(5) Fig.1 and 2 showed severe bias of the model in capturing the spatial pattern of precipitation distribution and vegetation distribution. Essentially, LAI has negligible difference between the Sahelian savannan and the central Africa forest. The discussion and statement about model performance in Section 3.1 significantly downplayed the severity of this model biases.

(6) The model biases in precipitation and more importantly in vegetation could significantly influence the location and magnitude of the difference between FB and NFB, and need to be discussed explicitly.

(7) Lines 315-320: The albedo difference is negligible? One would think that albedo changes can be significant in areas with increase of vegetation cover.

(8) Lines 448-453: This is not true. The state of the vegetation is very important in determining the interannual variability of vegetation and the vegetation feedback effects. This is why the issue of severe model bias needs to be acknowledged and its implication explicitly discussed, as suggested in comment 6).

Minor comments: Lines 92-94: " . . .. are important to . . ." is rather awkward. You mean " . . . are important determining factors for . . ." ? Fig. A4: "temperature gradient" should be changed to "temperature contrast" as y-label.

———————————————

---

## Author Comment (AC1) · 21 Apr 2016

Comments from reviewer #1:

This paper presents analysis of simulations with a regional climate model constrained by an earth system model (ESM) and coupled with a dynamic vegetation model (DVM). In the 21st c. simulation forced with the RCP8.5 scenario, including vegetation feedback led to drying in central Africa.

(1) A clarification on the model setup is needed. The authors mention (lines 478-481) that "SSTs were prescribed from CanESM2, therefore the land-ocean thermal contrast ... originated solely from the changes ... induced by vegetation dynamics". As far as I can see, this is the last mention of the prescribed SSTs in the paper, but exactly what are these prescribed SSTs? Climatology? For what period? Given the known

sensitive of the West and central African climate to SSTs, this needs to be explained carefully and the SSTs prescribed need to be evaluated. A coupled model such as the CanESM2 is not necessarily producing correct SSTs for the observational period. Of particular concern for the region of the analysis is the seasonal formation of the Atlantic cold tongue which, I believe, generally fails to form in coupled GCMs.

Response: The SSTs (section 2.2) are from the CanESM2 simulations and they are applied in the same manner as other boundary conditions, i.e., from the time-evolving GCM simulation. They are not a climatology (abbreviation "SST" will be added in line 167). Both the FB and the NFB simulation use the same set of SST forcing. The evaluation of CanESM2 SSTs in the oceans around Africa has been done in previous studies. Rowell (2013) indicated an acceptable agreement of CanESM2 SSTs with observations for their African teleconnection study; Xu et al. (2014) suggested relatively small SSTs biases from CanESM2 among CMIP5 models over the southeastern tropical Atlantic; LaRow et al. (2014) showed that SSTs of CanESM2 over the tropical oceans agree well with the reconstructed SSTs (derived from surface marine observational records). We will summarize the findings from these studies in the section 3.1 of the manuscript and will add references to these as "The SST forcing is also important for the African climate, and the CanESM2 SSTs have been validated and shown to be accurate in previous studies (e.g. Rowell, 2013;Xu et al., 2014;LaRow et al., 2014)"

(2)The biases in the regional model are significant (Fig. 1). The dry bias in the Congo Basin in the regional model (Fig. 1. b2), while common in models, seems extreme but it is similar to the dry bias in the ESM (Fig. 1.b3). It is important to consider how these biases influence the results, especially since one of the big results is additional drying in central Africa.

Response: The figure title "Model (CanESM2)" in the original version of Figure 1, referred to by the reviewer, refers to the NFB simulation with the RCM forced by CanESM2 boundary conditions, and not to the global CanESM2 model. We will clarified this by changing the title and figure text in Fig. 1. The dry bias is similar to other

RCMs (Kim et al., 2014;Nikulin et al., 2012), and very likely can be traced back to the convective scheme, rather than the circulation simulated by the physical sub-model. Following the reviewer's suggestion, we have also evaluated the low-level circulation and humidity in the CanESM-forced RCM simulation (the new figure Fig. A1), and we found that the dry bias over central Africa and wet bias over Sahelian savannah are not primarily related to the bias in circulation. In contrast, the model has done a relatively good job in reproducing the overall circulation patterns, including the southern and northern trade wind over the Atlantic ocean (the new figure Fig. A1) and the Walker circulation (Fig. 6), which is important for this study. We will give further explanations of this issue in section 3.1 & 4.3, and will add reference to the studies mentioned above.

(3)The wet bias in the Sahel in the regional model is unusual – many models fail to bring rainfall into the Sahel, as is the case for the ESM that is providing boundary conditions for the regional simulation (Fig. 1.b3). Is it relevant to the results that the regional model over-produces rainfall primarily in the spring?

Response: Yes, as shown by Fig.1b3, this could relate to the early onset of the rainy season. The issue is not unique to RCA and is common among RCMs (Kim et al., 2014). This can relate to the bias of the simulated West Africa Monsoon (WAM) dynamics, one possible explanation can be the biased interaction between deep convection and the Africa Easterly Waves (AEW). The propagation of AEW, which brings moisture to the Sahel regions, is dependent on the strength of deep convection: a strong deep convection can usually spread moisture at higher vertical atmospheric level, and cause rainfall over a wider latitudinal band along the ITCZ, whereas a weaker deep convection can result in a narrower but more concentrated precipitation band (Sylla et al., 2011). RCMs' sensitivity to the intensity of WAM can explain their different precipitation pattern over Sahel (Gbobaniyi et al., 2014). For the precipitation over central Africa, however, precipitation is primarily driven by orographic uplifting and low-level convergence, and it is maintained by low-level mass convergence over the ITCZ (Sylla et al., 2011) and the Walker circulation (Nicholson and Grist, 2003;Cook and Vizy, 2015). Therefore, the

influences from such bias on the dynamics in our study should be limited. We will give further explanations of this issue to section 3.1 as: "The simulated daily precipitation for central Africa tends to be underestimated during the late afternoon and night (Nikulin et al., 2012), resulting in dry bias. The wet bias over the northern savannah is mainly caused by a too early onset of the rainy season (b1, Fig. 2) which is possibly caused by the interactions between the simulated deep convection and the Africa Easterly Waves (Sylla et al., 2011)".

(4) I am puzzled by the large differences between the GPCP and CRU precipitation observations shown in the Sahel ("northern savanna") Fig. 2.b1. I think this is more related to the choice of averaging region than to a disparity in the observations, given the difference in the resolution of these 2 data sets. Please check this.

Response: We have investigated the possible errors of the interpolation from a coarser to a finer resolution. In our study we followed the approach in Nikulin et al. (2012) with bilinear interpolation for the GPCP (1 degree) and CRU datasets (0.5 degree) to regrid respective data to 0.44 degree. We have performed some technical checks on our analysis, including a check for mass conservation, and found mass differences due to the regridding negligible, difference is less than 0.01%. The disparities between the observation-based precipitation datasets are not unique. The CRU dataset is particularly challenged by the small amount of observing stations in Africa, and there of course are other well-recognised issues with other datasets, including satellite-based, which cause differences. Examples of studies that have commented on this are Nikulin et al. (2012) and Kim et al. (2014). They find considerable disparities among observation-based precipitation products, including TRMM and UDEL, also found for the Sahel region. We will give further information to section 3.1.

(5)References to the Charney (1975) and related studies are problematic since the idea that vegetation changes (i.e., "over-grazing") caused the precipitation decline in West Africa during the 1960's and 70's has been thoroughly refuted in the mode modern literature. It's SSTs forcing, of course.

Response: We agree that the role of SSTs is central, but also want to acknowledge Charney's paper which was seminal in hypothesising the potential impacts of vegetation changes on the monsoon circulation, which has formed the basis for many other vegetation change-related studies, not least those related to long-term vegetation changes in this region. Our study investigates how vegetation changes can lead to feedback in the region. This will be rephrased to "Hypothesised mechanisms of vegetation-atmosphere coupling include modulations of the surface albedo (Charney, 1975), changes in the North-African monsoon system (Claussen, 1997) and internal climate variability (Zeng et al., 1999)." in the revised manuscript.

(6)The authors note (lines 240-242) that "The simulated patterns and magnitude of precipitation for this area are similar to a previous study using an earlier version of RCA, RCA3.5, without dynamic vegetation". So doesn't that mean that dynamic vegetation is not influential, in contrast to the findings of this paper?

Response: The comparison to RCA3.5 in previous study refers to the simulated present-day climate. For the present-day period, influences from vegetation dynamics are limited as the present-day land cover types in terms of forest cover and open land are able to reproduce (Fig. 4a) though given the bias in LAI, and large-scale vegetation changes rarely happen over the short period of comparison. However, for the century-long transition period under climate change considered in this paper, changes in climate and CO2 forcing are strong enough, and lag effects of vegetation response short enough, to induce large-scale and long-term vegetation change and its feedback effect on climate is found to be much stronger than that seen during the present-day period. This will be further explained in section 4.3 as "Despite biases in the initial precipitation and vegetation state (LAI) for some regions, our model was able to reproduce the present-day land cover type in terms of forest cover and open land (Fig. 4a), and the simulated present-date climate is close to previous study (Nikulin et al., 2012) using the same physical sub-model with observed land cover type. Under future climate change, vegetation-induced changes in circulation, thus a substantial change

in moisture transport and precipitation, are mainly triggered by changes in land cover type (Fig. 4A), therefore, we argue that the influences from biases in initial conditions on such mechanism found in this study should be limited."

(7)I would appreciate seeing an evaluation (e.g., a comparison with the ERAI reanalysis) of the circulation and specific humidity at 850 hPa wind and specific humidity from the present day, NFB simulation since the authors are pointing to changes in the circulation/moisture advection as relevant. This seems more crucial than evaluating LAI, for example.

Response: Thank you for this suggestion. A new figure will be added (as Fig. A1) in the Appendix and new text will be added in the section 3.1. and Fig. A1 can be found in the attached Supplement file in the comment system.

(8)There's not a lot of literature on the dynamics of the Walker circulation in this region and its sensitivity to SSTs (and/or land/sea contrast), but these recent papers will help: Pokam WM, Djiotang LAT, Mkankam FK, 2012: Atmospheric water vapor transport and recycling in equatorial central Africa through NCEP/NCAR reanalysis data. Climate Dyn. 38, 1715-1729. Pokam MW, Bain CL, Chadwick RS, Graham R, Sonwa DJ, Kamga FM, 2014: Identification of processes driving low-level westerlies in West Equatorial Africa. J. Climate 27, 4245-4262. Cook, K. H., and E. K. Vizy, 2015: The Congo Basin Walker Circulation: Dynamics and Connections to Precipitation, Climate Dynamics, DOI 10,1007/s00382-015-2864-y.

Response: Thank you for pointing us to these studies, new references will be added to the section 3.3 and 3.4.

(9)A couple of minor points: Please note that "Savannah" is the city in Georgia, U.S., while "savanna" is the grassland.

Response: The Oxford English Dictionary gives "savannah" as the preferred spelling.

(10)Figure A1 caption needs to be improved to provide more detail about what is plotted.

Response: Agreed. The revised caption will appears as "Fig. A2. Simulated seasonal surface temperature for present day (a-d), for changes in future in the NFB experiment (e-h, future minus present day), and for changes from vegetation feedback in future (i-l, FB minus NFB for future). Definitions for calculation period, climate change signal and vegetation feedbacks are given in Sect. 2.2."

Please also note the supplement to this comment:
http://www.earth-syst-dynam-discuss.net/esd-2015-88/esd-2015-88-AC1-supplement.pdf

**Supplement:**

Dear Prof. Dr. Claussen, dear reviewers,

Thank you very much for your assessment and the constructive comments to improve our manuscript. Our response is detailed below, as well as being reflected in the revised manuscript. We hope that we have addressed all concerns and questions that were raised.

With best regards,

Minchao Wu (on behalf of all authors)

**Comments from reviewer #1:**

This paper presents analysis of simulations with a regional climate model constrained by an earth system model (ESM) and coupled with a dynamic vegetation model (DVM). In the 21st c. simulation forced with the RCP8.5 scenario, including vegetation feedback led to drying in central Africa.

(1) A clarification on the model setup is needed. The authors mention (lines 478-481) that "SSTs were prescribed from CanESM2, therefore the land-ocean thermal contrast … originated solely from the changes … induced by vegetation dynamics". As far as I can see, this is the last mention of the prescribed SSTs in the paper, but exactly what are these prescribed SSTs? Climatology? For what period? Given the known sensitive of the West and central African climate to SSTs, this needs to be explained carefully and the SSTs prescribed need to be evaluated. A coupled model such as the CanESM2 is not necessarily producing correct SSTs for the observational period. Of particular concern for the region of the analysis is the seasonal formation of the Atlantic cold tongue which, I believe, generally fails to form in coupled GCMs.

Response: The SSTs (section 2.2) are from the CanESM2 simulations and they are applied in the same manner as other boundary conditions, i.e., from the time-evolving GCM simulation. They are not a climatology (abbreviation "SST" will be added in line 167). Both the FB and the NFB simulation use the same set of SST forcing.

The evaluation of CanESM2 SSTs in the oceans around Africa has been done in previous studies. Rowell (2013) indicated an acceptable agreement of CanESM2 SSTs with observations for their African teleconnection study; Xu et al. (2014) suggested relatively small SSTs biases from CanESM2 among CMIP5 models over the southeastern tropical Atlantic; LaRow et al. (2014) showed that SSTs of CanESM2 over the tropical oceans agree well with the reconstructed SSTs (derived from surface marine observational records).

We will summarize the findings from these studies in the section 3.1 of the manuscript and will add references to these as "The SST forcing is also important for the African climate, and the CanESM2 SSTs have been validated and shown to be accurate in previous studies (e.g. Rowell, 2013;Xu et al., 2014;LaRow et al., 2014)"

(2)The biases in the regional model are significant (Fig. 1). The dry bias in the Congo Basin in the regional model (Fig. 1. b2), while common in models, seems extreme but it is similar to the dry bias in the ESM (Fig. 1.b3). It is important to consider how these biases influence the results, especially since one of the big results is additional drying in central Africa.

Response: The figure title "Model (CanESM2)" in the original version of Figure 1, referred to by the reviewer, refers to the NFB simulation with the RCM forced by CanESM2 boundary conditions, and not to the global CanESM2 model. We will clarified this by changing the title and figure text in Fig. 1.

The dry bias is similar to other RCMs (Kim et al., 2014;Nikulin et al., 2012), and very likely can be traced back to the convective scheme, rather than the circulation simulated by the physical sub-model. Following the reviewer's suggestion, we have also evaluated the low-level circulation and humidity in the CanESM-forced RCM simulation (the new figure Fig. A1), and we found that the dry bias over central Africa and wet bias over Sahelian savannah are not primarily related to the bias in circulation. In contrast, the model has done a relatively good job in reproducing the overall circulation patterns, including the southern and northern trade wind over the Atlantic ocean (the new figure Fig. A1) and the Walker circulation (Fig. 6), which is important for this study.

We will give further explanations of this issue in section 3.1 & 4.3, and will add reference to the studies mentioned above.

(3)The wet bias in the Sahel in the regional model is unusual – many models fail to bring rainfall into the Sahel, as is the case for the ESM that is providing boundary conditions for the regional simulation (Fig. 1.b3). Is it relevant to the results that the regional model over-produces rainfall primarily in the spring?

Response: Yes, as shown by Fig.1b3, this could relate to the early onset of the rainy season. The issue is not unique to RCA and is common among RCMs (Kim et al., 2014). This can relate to the bias of the simulated West Africa Monsoon (WAM) dynamics, one possible explanation can be the biased interaction between deep convection and the Africa Easterly Waves (AEW). The propagation of AEW, which brings moisture to the Sahel regions, is dependent on the strength of deep convection: a strong deep convection can usually spread moisture at higher vertical atmospheric  level, and cause rainfall over a wider latitudinal band along the ITCZ, whereas a weaker deep convection can result in a narrower but more concentrated precipitation band (Sylla et al., 2011). RCMs' sensitivity to the intensity of WAM can explain their different precipitation pattern over Sahel (Gbobaniyi et al., 2014). For the precipitation over central Africa, however, precipitation is primarily driven by orographic uplifting and low-level convergence, and it is maintained by low-level mass convergence over the ITCZ (Sylla et al., 2011) and the

Walker circulation (Nicholson and Grist, 2003;Cook and Vizy, 2015). Therefore, the influences from such bias on the dynamics in our study should be limited.

We will give further explanations of this issue to section 3.1 as: "The simulated daily precipitation for central Africa tends to be underestimated during the late afternoon and night (Nikulin et al., 2012), resulting in dry bias. The wet bias over the northern savannah is mainly caused by a too early onset of the rainy season (b1, Fig. 2) which is possibly caused by the interactions between the simulated deep convection and the Africa Easterly Waves (Sylla et al., 2011)".

(4) I am puzzled by the large differences between the GPCP and CRU precipitation observations shown in the Sahel ("northern savanna") Fig. 2.b1. I think this is more related to the choice of averaging region than to a disparity in the observations, given the difference in the resolution of these 2 data sets. Please check this.

Response:  We have investigated the possible errors of the interpolation from a coarser to a finer resolution. In our study we followed the approach in Nikulin et al. (2012) with bilinear interpolation for the GPCP (1 degree) and CRU datasets (0.5 degree) to regrid respective data to 0.44 degree. We have performed some technical checks on our analysis, including a check for mass conservation, and found mass differences due to the regridding negligible, difference is less than 0.01%.

The disparities between the observation-based precipitation datasets are not unique. The CRU dataset is particularly challenged by the small amount of observing stations in Africa, and there of course are other well-recognised issues with other datasets, including satellite-based, which cause differences. Examples of studies that have commented on this are Nikulin et al. (2012) and Kim et al. (2014). They find considerable disparities among observation-based precipitation products, including TRMM and UDEL, also found for the Sahel region.

We will give further information to section 3.1.

(5)References to the Charney (1975) and related studies are problematic since the idea that vegetation changes (i.e., "over-grazing") caused the precipitation decline in West Africa during the 1960's and 70's has been thoroughly refuted in the mode modern literature.  It's SSTs forcing, of course.

Response:  We agree that the role of SSTs is central, but also want to acknowledge Charney's paper which was seminal in hypothesising the potential impacts of vegetation changes on the monsoon circulation, which has formed the basis for many other vegetation change-related studies, not least those related to long-term vegetation changes in this region. Our study investigates how vegetation changes can lead to feedback in the region.

This will be rephrased to "Hypothesised mechanisms of vegetation-atmosphere coupling include modulations of the surface albedo (Charney, 1975), changes in the North-African monsoon system (Claussen, 1997) and internal climate variability (Zeng et al., 1999)." in the revised manuscript.

(6)The authors note (lines 240-242) that "The simulated patterns and magnitude of precipitation for this area are similar to a previous study using an earlier version of RCA, RCA3.5, without dynamic vegetation". So doesn't that mean that dynamic vegetation is not influential, in contrast to the findings of this paper?

Response:  The comparison to RCA3.5 in previous study refers to the simulated present-day climate. For the present-day period, influences from vegetation dynamics are limited as the present-day land cover types in terms of forest cover and open land are able to reproduce (Fig. 4a) though given the bias in LAI, and large-scale vegetation changes rarely happen over the short period of comparison. However, for the century-long transition period under climate change considered in this paper, changes in climate and $CO_2$ forcing are strong enough, and lag effects of vegetation response short enough, to induce large-scale and long-term vegetation change and its feedback effect on climate is found to be much stronger than that seen during the present-day period.

This will be further explained in section 4.3 as "Despite biases in the initial precipitation and vegetation state (LAI) for some regions, our model was able to reproduce the present-day land cover type in terms of forest cover and open land (Fig. 4a), and the simulated present-date climate is close to previous study (Nikulin et al., 2012) using the same physical sub-model with observed land cover type. Under future climate change, vegetation-induced changes in circulation, thus a substantial change in moisture transport and precipitation, are mainly triggered by changes in land cover type (Fig. 4A), therefore, we argue that the influences from biases in initial conditions on such mechanism found in this study should be limited."

(7)I would appreciate seeing an evaluation (e.g., a comparison with the ERAI reanalysis) of the circulation and specific humidity at 850 hPa wind and specific humidity from the present day, NFB simulation since the authors are pointing to changes in the circulation/moisture advection as relevant. This seems more crucial than evaluating LAI, for example.

Response:  Thank you for this suggestion. A new figure will be added (as Fig. A1) in the Appendix and new text will be added in the section 3.1.

(8)There's not a lot of literature on the dynamics of the Walker circulation in this region and its sensitivity to SSTs (and/or land/sea contrast), but these recent papers will help: Pokam WM, Djiotang LAT, Mkankam FK, 2012: Atmospheric water vapor transport and

recycling in equatorial central Africa through NCEP/NCAR reanalysis data. Climate
Dyn. 38, 1715-1729.

Pokam MW, Bain CL, Chadwick RS, Graham R, Sonwa DJ, Kamga FM, 2014: Identification
of processes driving low-level westerlies in West Equatorial Africa. J. Climate
27, 4245-4262.

Cook, K. H., and E. K. Vizy, 2015: The Congo Basin Walker Circulation: Dynamics and
Connections to Precipitation, Climate Dynamics, DOI 10,1007/s00382-015-2864-y.

Response: Thank you for pointing us to these studies, new references will be added to the
section 3.3 and 3.4.

(9)A couple of minor points:

Please note that "Savannah" is the city in Georgia, U.S., while "savanna" is the grassland.

Response: The Oxford English Dictionary gives "savannah" as the preferred spelling.

 (10)Figure A1 caption needs to be improved to provide more detail about what is plotted.

Response:  Agreed. The revised caption will appears as "Fig. A2. Simulated seasonal surface
temperature for present day (a-d), for changes in future in the NFB experiment (e-h, future
minus present day), and for changes from vegetation feedback in future (i-l, FB minus NFB for
future). Definitions for calculation period, climate change signal and vegetation feedbacks are
given in Sect. 2.2."

**Comments from reviewer #2:**

This paper presents a future prediction study on climate-vegetation interactions in Africa. While the concept is not new, it does add to an emerging body of literature on interactive vegetation-climate predictions and will be of interest to many readers of ESD. The paper potentially merits publication, but quite a few major issues need to be addressed:

(1) Introduction: The flow of thought is very hard to follow. Part of the reason has to do with a rather liberal use of terminology. Probably a more strict use of the words "change" "variability" "pattern" "feedback" will help. The way it is now, many sentences are either vague or not accurate, which does not serve the readers well. Needs a better organization.

Response: We will revise the introduction and will attempt to use a more strict terminology throughout the article.

(2) Introduction: An important body of literature (e.g., Claussen 1997 climate dynamics, Claussen 1998 global change biology; Zeng et al., 1999 science; Alo & Wang, 2010 climate dyamics; Yu et al., 2015 climate dynamics ) on vegetation-climate interactions is missing, although some of them are later mentioned in the Discussion section. The introduction part of a paper should be the place where the status of science is conveyed and gaps identified. Otherwise it will be misleading for readers who are new to the topic.

Response: We will add a new paragraph to the introduction where we will cite most of these highly relevant papers and highlight the specific gaps that our paper is attempting to address.

(3) Partly related to (2), the statement in lines 111-112 is misleading. The first several sentences in section 4.1 should be moved here to provide readers an accurate description of the status of science, and the authors need to further elaborate to explain why this study adds values to existing literature.

Response: This will be addressed, see response to previous comment. The revised statement will be : "Recent studies have used a regional climate model to investigate the impact of climate-vegetation interaction for West Africa, identifying significant vegetation feedback in modulating local hydrological cycling (e.g. Wang and Alo, 2012;Yu et al., 2015;Alo and Wang, 2010). Additionally, a number of GCM-based studies have investigated the climate effects of anthropogenic perturbations, such as deforestation or afforestation (e.g. Lawrence and Vandecar, 2015). Such studies point to potentially significant forcing of regional climate dynamics, particularly rainfall patterns, as a result of changes in land cover. No study to date has, however, characterised the coupled dynamics of vegetation and climate under future radiative forcing for the entire African domain at a grid resolution high enough to capture regional features and forcings."

(4) Fig.1: The color scale is very difficult to read if one were to try to figure out the actual magnitude of the model biases. Should use more distinguishable color scales/ use stronger contrast between the colors.

Response: This will be addressed, thanks for the suggestion.

(5) Fig.1 and 2 showed severe bias of the model in capturing the spatial pattern of precipitation distribution and vegetation distribution. Essentially, LAI has negligible difference between the Sahelian savannan and the central Africa forest. The discussion and statement about model performance in Section 3.1 significantly downplayed the severity of this model biases.

Response: We agree that these biases are significant. We will add a substantial discussion of the bias in precipitation pattern and LAI in Section 3.1, such as acknowledgement for the LAI bias "A systematic overestimation is apparent for savannahs, and a significant underestimation for the central Africa rainforest area" and more explanation for the precipitation bias"The simulated daily precipitation for central Africa tends to be underestimated during the late afternoon and night (Nikulin et al., 2012), resulting in dry bias. The wet bias over the northern savannah is mainly caused by a too early onset of the rainy season (b1, Fig. 2) which is possibly caused by the interactions between the simulated deep convection and the Africa Easterly Waves (Sylla et al., 2011)"

Additionally, in response to Comment (7) from Reviewer #1 which is also relevant to this comments, we will add a new figure (Fig A1) and related discussion evaluating low-level circulation and humidity. We found that the dry bias over central Africa and wet bias over Sahelian savannah are not primarily related to the biases in the circulation, but are more likely to be related to problems with the convection scheme in the regional model (Nikulin et al. 2012). In contrast, the model has done a relatively good job in reproducing overall circulation patterns, including the southern and northern trade wind over Atlantic oceans (Fig. A1), Walker circulation (Fig. 6), which are important for this study. This will be included in section 3.1 also.

(6) The model biases in precipitation and more importantly in vegetation could significantly influence the location and magnitude of the difference between FB and NFB, and need to be discussed explicitly.

Response: For the vegetation dynamics, the bias in simulated present-day vegetation is largely related to a bias in precipitation. We agree that this potentially influences the simulated difference between FB and NFB, as the latter uses the resultant, biased vegetation as forcing. This may lead to an offset in the locations of the strongest impact of vegetation feedbacks in the model, but we assume that this bias does not critically affect qualitative aspects of the

feedbacks that we find. One reason for such confidence is that the bias is small in magnitude compared to the size of the simulated future changes in LAI and precipitation. We will add more discussion in section 4.2.

The influence of the bias on model's dynamics is explained in point (8) further below.

(7) Lines 315-320: The albedo difference is negligible? One would think that albedo changes can be significant in areas with increase of vegetation cover.

Response: We agree that albedo changes play a role for surface temperature changes, this was ill-phrased in the original manuscript. We have identified warming effects from albedo change, which gives an overall warming effect in northern hemisphere winter on the edge of the area of forest expansion in the northern savannah region (Fig. A2). An increase in vegetation (forest) cover gives both an albedo (warming) effect and an evaporative (cooling) effect, with the combined effect depending on their seasonal balance. In general, modelling studies tend to show that evaporative cooling effects are more dominant in the tropics while albedo warming effects are more dominant over high-latitude regions (e.g. Bala et al., 2007;Claussen et al., 2001).

The missing explanation for the albedo effect will be added to section 3.3 as "Overall, the turbulent heat fluxes increase, which tends to cool the surface and the lower atmosphere, exceeding the opposing (warming) effects of increased vegetation cover on albedo, thus resulting in an overall cooling effect.".

(8) Lines 448-453: This is not true. The state of the vegetation is very important in determining the interannual variability of vegetation and the vegetation feedback effects. This is why the issue of severe model bias needs to be acknowledged and its implication explicitly discussed, as suggested in comment 6).

Response: We agree with the reviewer on this point, we did not express our point well in the original manuscript. We intended to point out that bias in LAI within a given land cover type (forest, savannah or grassland) is likely to have a smaller impact on the simulated climate than an inaccurate distribution of land cover types. Although our simulations have evident bias in LAI and precipitation, overall patterns of vegetation distribution across Africa are comparable to observations.

This will be further explained in section 4.2 as "Despite biases in the initial precipitation and vegetation state (LAI) for some regions, our model was able to reproduce the present-day land cover type (Fig. 4a). Vegetation-induced changes in circulation, thus a substantial change in moisture transport and precipitation, are mainly triggered by changes in land cover type (Fig. 4A), therefore, we argue that the influences from biases in initial conditions on such mechanism found in this study should be limited."

Minor comments: Lines 92-94: "… are important to … " is rather awkward. You mean
"… are important determining factors for …" ? Fig. A4: "temperature gradient" should
be changed to "temperature contrast" as y-label.

Response: Will change as suggested, thank you for the suggestions.

[Figure]

Fig. A1. Seasonal atmospheric circulation (arrows, m s$^{-1}$) and specific humidity (colour contours, g kg$^{-1}$) at
850 hPa pressure level from ERA-Interim (1$^{st}$ row), NFB run (2$^{nd}$ row), as well as their differences (3$^{rd}$ row,
NFB minus ERA-Interim), for the period 1997-2010.

**Reference:**

Alo, C. A., and Wang, G.: Role of dynamic vegetation in regional climate predictions over western Africa, Climate dynamics, 35, 907-922, 2010.

Bala, G., Caldeira, K., Wickett, M., Phillips, T., Lobell, D., Delire, C., and Mirin, A.: Combined climate and carbon-cycle effects of large-scale deforestation, Proceedings of the National Academy of Sciences, 104, 6550-6555, 2007.

Charney, J. G.: Dynamics of deserts and drought in the Sahel, Quarterly Journal of the Royal Meteorological Society, 101, 193-202, 1975.

Claussen, M.: Modeling bio-geophysical feedback in the African and Indian monsoon region, Climate Dynamics, 13, 247-257, 1997.

Claussen, M., Brovkin, V., and Ganopolski, A.: Biogeophysical versus biogeochemical feedbacks of large‐scale land cover change, Geophysical research letters, 28, 1011-1014, 2001.

Cook, K. H., and Vizy, E. K.: The Congo Basin Walker circulation: dynamics and connections to precipitation, Climate Dynamics, 1-21, 2015.

Gbobaniyi, E., Sarr, A., Sylla, M. B., Diallo, I., Lennard, C., Dosio, A., Dhiédiou, A., Kamga, A., Klutse, N. A. B., and Hewitson, B.: Climatology, annual cycle and interannual variability of precipitation and temperature in CORDEX simulations over West Africa, International Journal of Climatology, 34, 2241-2257, 2014.

Kim, J., Waliser, D. E., Mattmann, C. A., Goodale, C. E., Hart, A. F., Zimdars, P. A., Crichton, D. J., Jones, C., Nikulin, G., and Hewitson, B.: Evaluation of the CORDEX-Africa multi-RCM hindcast: systematic model errors, Climate dynamics, 42, 1189-1202, 2014.

LaRow, T. E., Stefanova, L., and Seitz, C.: Dynamical simulations of north Atlantic tropical cyclone activity using observed low-frequency SST oscillation imposed on CMIP5 Model RCP4. 5 SST projections, Journal of Climate, 27, 8055-8069, 2014.

Lawrence, D., and Vandecar, K.: Effects of tropical deforestation on climate and agriculture, Nature Climate Change, 5, 27-36, 2015.

Nicholson, S. E., and Grist, J. P.: The seasonal evolution of the atmospheric circulation over West Africa and equatorial Africa, Journal of Climate, 16, 1013-1030, 2003.

Nikulin, G., Jones, C., Giorgi, F., Asrar, G., Büchner, M., Cerezo-Mota, R., Christensen, O. B., Déqué, M., Fernandez, J., Hänsler, A., van Meijgaard, E., Samuelsson, P., Sylla, M. B., and Sushama, L.: Precipitation Climatology in an Ensemble of CORDEX-Africa Regional Climate Simulations, Journal of Climate, 25, 6057-6078, 10.1175/JCLI-D-11-00375.1, 2012.

Rowell, D. P.: Simulating SST teleconnections to Africa: What is the state of the art?, Journal of Climate, 26, 5397-5418, 2013.

Sylla, M., Giorgi, F., Ruti, P., Calmanti, S., and Dell'Aquila, A.: The impact of deep convection on the West African summer monsoon climate: a regional climate model sensitivity study, Quarterly Journal of the Royal Meteorological Society, 137, 1417-1430, 2011.

Wang, G., and Alo, C. A.: Changes in precipitation seasonality in West Africa predicted by RegCM3 and the impact of dynamic vegetation feedback, International Journal of Geophysics, 2012, 2012.

Xu, Z., Chang, P., Richter, I., and Tang, G.: Diagnosing southeast tropical Atlantic SST and ocean circulation biases in the CMIP5 ensemble, Climate dynamics, 43, 3123-3145, 2014.

Yu, M., Wang, G., and Pal, J. S.: Effects of vegetation feedback on future climate change over West Africa, Climate Dynamics, 1-20, 2015.

Zeng, N., Neelin, J. D., Lau, K.-M., and Tucker, C. J.: Enhancement of interdecadal climate variability in the Sahel by vegetation interaction, Science, 286, 1537-1540, 1999.

---

## Author Response (AR1)

Dear Prof. Dr. Claussen, dear reviewers,

Thank you very much for your assessment and the constructive comments to improve our manuscript. Our response is detailed below, as well as being reflected in the revised manuscript. We hope that we have addressed all concerns and questions that were raised.

With best regards,

Minchao Wu (on behalf of all authors)

**Comments from reviewer #1:**

This paper presents analysis of simulations with a regional climate model constrained by an earth system model (ESM) and coupled with a dynamic vegetation model (DVM). In the 21st c. simulation forced with the RCP8.5 scenario, including vegetation feedback led to drying in central Africa.

(1) A clarification on the model setup is needed. The authors mention (lines 478-481) that "SSTs were prescribed from CanESM2, therefore the land-ocean thermal contrast … originated solely from the changes … induced by vegetation dynamics". As far as I can see, this is the last mention of the prescribed SSTs in the paper, but exactly what are these prescribed SSTs? Climatology? For what period? Given the known sensitive of the West and central African climate to SSTs, this needs to be explained carefully and the SSTs prescribed need to be evaluated. A coupled model such as the CanESM2 is not necessarily producing correct SSTs for the observational period. Of particular concern for the region of the analysis is the seasonal formation of the Atlantic cold tongue which, I believe, generally fails to form in coupled GCMs.

Response: The SSTs (section 2.2) are from the CanESM2 simulations and they are applied in the same manner as other boundary conditions, i.e., from the time-evolving GCM simulation. They are not a climatology (abbreviation "SST" has been added in lines 172 and 176 in the revised manuscript). Both the FB and the NFB simulation use the same set of SST forcing.

The evaluation of CanESM2 SSTs in the oceans around Africa has been done in previous studies. Rowell (2013) indicated an acceptable agreement of CanESM2 SSTs with observations for their African teleconnection study; Xu et al. (2014) suggested relatively small SSTs biases from CanESM2 among CMIP5 models over the southeastern tropical Atlantic; LaRow et al. (2014) showed that SSTs of CanESM2 over the tropical oceans agree well with the reconstructed SSTs (derived from surface marine observational records).

We has summarised the findings from these studies in the section 3.1 of the revised manuscript (lines 265-267), and relevant references has been added,  the revised texts are as "The SST forcing is also important for the African climate, and the CanESM2 SSTs have been validated and shown to be accurate in previous studies (e.g. Rowell, 2013;LaRow et al., 2014;Xu et al., 2014)".

(2)The biases in the regional model are significant (Fig. 1). The dry bias in the Congo Basin in the regional model (Fig. 1. b2), while common in models, seems extreme but it is similar to the dry bias in the ESM (Fig. 1.b3). It is important to consider how these biases influence the results, especially since one of the big results is additional drying in central Africa.

Response: The figure title "Model (CanESM2)" in the original version of Figure 1, referred to by the reviewer, refers to the NFB simulation with the RCM forced by CanESM2 boundary conditions, and not to the global CanESM2 model. We have clarified this by changing the title and figure text in Fig. 1.

On the other hand, there may be an issue for CRU dataset conversion applied in our post-processing for model evaluation. We have now used the consistent conversion method and found that this can reduce model bias for Sahel and central Africa, and reduce observation uncertainty, more information can be found in comments #4. The dry bias is similar to other RCMs (Nikulin et al., 2012;Kim et al., 2014), and very likely can be traced back to the convective scheme, rather than the circulation simulated by the physical sub-model. Following the reviewer's suggestion, we have also evaluated the low-level circulation and humidity in the CanESM-forced RCM simulation (the new figure Fig. A1), and we found that the dry bias over central Africa and wet bias over Sahelian savannah are not primarily related to the bias in circulation. In contrast, the model has done a relatively good job in reproducing the overall circulation patterns, including the southern and northern trade wind over the Atlantic ocean (the new figure Fig. A1) and the Walker circulation (Fig. 6), which is important for this study.

We have given further explanations of this issue in section 3.1 & 4.3 (lines 247 – 277, lines 512 – 523 in the revised manuscript), and relevant references has been added accordingly.

(3)The wet bias in the Sahel in the regional model is unusual – many models fail to bring rainfall into the Sahel, as is the case for the ESM that is providing boundary conditions for the regional simulation (Fig. 1.b3). Is it relevant to the results that the regional model over-produces rainfall primarily in the spring?

Response:  Yes, as shown by Fig.1b3, this could relate to the early onset of the rainy season. The issue is not unique to RCA and is common among RCMs (Kim et al., 2014). This can relate to the bias of the simulated West Africa Monsoon (WAM) dynamics, one possible explanation can be the biased interaction between deep convection and the Africa Easterly Waves (AEW). The propagation of AEW, which brings moisture to the Sahel regions, is dependent on the strength of deep convection: a strong deep convection can usually spread moisture at higher vertical atmospheric  level, and cause rainfall over a wider latitudinal band along the ITCZ, whereas a weaker deep convection can result in a narrower but more concentrated precipitation band (Sylla et al., 2011). RCMs' sensitivity to the intensity of WAM can explain their different precipitation pattern over Sahel (Gbobaniyi et al., 2014). For the precipitation over central Africa, however, precipitation is primarily driven by orographic uplifting and low-level convergence, and it is maintained by low-level mass convergence over the ITCZ (Sylla et al., 2011) and the

Walker circulation (Nicholson and Grist, 2003;Cook and Vizy, 2015). Therefore, the influences from such bias on the dynamics in our study should be limited.

We have given further explanations of this issue to section 3.1 (lines 247-253 in the revised manuscript) as: "the dry bias for annual mean precipitation over central Africa may be partly due to the underestimated daily precipitation during the late afternoon and night in addition to observational uncertainties (Nikulin et al., 2012). The wet bias over the northern savannah is mainly caused by a too early onset of the rainy season (b1, Fig. 2), which is possibly caused by the interactions between the simulated deep convection and the Africa Easterly Waves (Sylla et al., 2011)".

(4) I am puzzled by the large differences between the GPCP and CRU precipitation observations shown in the Sahel ("northern savanna") Fig. 2.b1. I think this is more related to the choice of averaging region than to a disparity in the observations, given the difference in the resolution of these 2 data sets. Please check this.

Response: Thanks for spotting this for us, this may relate to the inconsistent data conversion methods applied in our data post-processing for CRU dataset and GPCP in this study, have now corrected this with the latest CRU dataset TS3.23, we found that model bias and observation uncertainty have reduced. This has been updated accordingly in section 3.1 (line 220, lines 244-245 in the revised manuscript, Fig 1&2 are now updated with the new CRU dataset).

(5)References to the Charney (1975) and related studies are problematic since the idea that vegetation changes (i.e., "over-grazing") caused the precipitation decline in West Africa during the 1960's and 70's has been thoroughly refuted in the mode modern literature. It's SSTs forcing, of course.

Response: We agree that the role of SSTs is central, but also want to acknowledge Charney's paper which was seminal in hypothesising the potential impacts of vegetation changes on the monsoon circulation, which has formed the basis for many other vegetation change-related studies, not least those related to long-term vegetation changes in this region. Our study investigates how vegetation changes can lead to feedback in this region, changes in albedo, land surface temperature, thus land ocean contrast is relevant to this hypothesis.

Here we would like to rephrase in lines 103-106 in the revised manuscript to "Hypothesised mechanisms of vegetation-atmosphere coupling include modulations of the surface albedo (Charney, 1975), changes in the North-African monsoon system (Claussen, 1997) and internal climate variability (Zeng et al., 1999)".

(6)The authors note (lines 240-242) that "The simulated patterns and magnitude of precipitation for this area are similar to a previous study using an earlier version of RCA, RCA3.5, without dynamic vegetation". So doesn't that mean that dynamic vegetation is not influential, in contrast to the findings of this paper?

Response:  The comparison to RCA3.5 in previous study refers to the simulated present-day climate. For the present-day period, influences from vegetation dynamics are limited as the present-day land cover types in terms of forest cover and open land are able to reproduce though given the bias in LAI, and large-scale vegetation changes rarely happen over the short period of comparison. However, for the century-long transition period under climate change considered in this paper, changes in climate and $CO_2$ forcing are strong enough, and lag effects of vegetation response short enough, to induce large-scale and long-term vegetation change and its feedback effect on climate is found to be much stronger than that seen during the present-day period.

(7)I would appreciate seeing an evaluation (e.g., a comparison with the ERAI reanalysis) of the circulation and specific humidity at 850 hPa wind and specific humidity from the present day, NFB simulation since the authors are pointing to changes in the circulation/moisture advection as relevant. This seems more crucial than evaluating LAI, for example.

Response:  Thank you for this suggestion. A new figure has been added (as Fig. A1 in the revised manuscript) in the Appendix and new text has been added in the section 3.1 (lines 263-277).

(8)There's not a lot of literature on the dynamics of the Walker circulation in this region and its sensitivity to SSTs (and/or land/sea contrast), but these recent papers will help:
Pokam WM, Djiotang LAT, Mkankam FK, 2012: Atmospheric water vapor transport and recycling in equatorial central Africa through NCEP/NCAR reanalysis data. Climate Dyn. 38, 1715-1729.
Pokam MW, Bain CL, Chadwick RS, Graham R, Sonwa DJ, Kamga FM, 2014: Identification of processes driving low-level westerlies in West Equatorial Africa. J. Climate 27, 4245-4262.
Cook, K. H., and E. K. Vizy, 2015: The Congo Basin Walker Circulation: Dynamics and Connections to Precipitation, Climate Dynamics, DOI 10,1007/s00382-015-2864-y.

Response: Thank you for pointing us to these studies, new references has been added to the section 3.3 (line 349) and section 3.4 (line 378).

(9)A couple of minor points:
Please note that "Savannah" is the city in Georgia, U.S., while "savanna" is the grassland.
Response: The Oxford English Dictionary gives "savannah" as the preferred spelling.

 (10)Figure A1 caption needs to be improved to provide more detail about what is plotted.
Response:  Agreed. The revised caption now appears as "Fig. A2. Simulated seasonal surface temperature for present day (a-d), for changes in future in the NFB experiment (e-h, future minus present day), and for changes from vegetation feedback in future (i-l, FB minus NFB for future). Definitions for calculation period, climate change signal and vegetation feedbacks are given in Sect. 2.2."

**Comments from reviewer #2:**

This paper presents a future prediction study on climate-vegetation interactions in Africa. While the concept is not new, it does add to an emerging body of literature on interactive vegetation-climate predictions and will be of interest to many readers of ESD. The paper potentially merits publication, but quite a few major issues need to be addressed:

(1) Introduction: The flow of thought is very hard to follow. Part of the reason has to do with a rather liberal use of terminology. Probably a more strict use of the words "change" "variability" "pattern" "feedback" will help. The way it is now, many sentences are either vague or not accurate, which does not serve the readers well. Needs a better organization.

Response: We have revised the introduction and have attempted to use a more strict terminology throughout the article.

(2) Introduction: An important body of literature (e.g., Claussen 1997 climate dynamics, Claussen 1998 global change biology; Zeng et al., 1999 science; Alo & Wang, 2010 climate dyamics; Yu et al., 2015 climate dynamics ) on vegetation-climate interactions is missing, although some of them are later mentioned in the Discussion section. The introduction part of a paper should be the place where the status of science is conveyed and gaps identified. Otherwise it will be misleading for readers who are new to the topic.

Response: We have added new texts to the introduction (lines 101-119) where we have cited most of these highly relevant papers and highlight the specific gaps that our paper is attempting to address.

(3) Partly related to (2), the statement in lines 111-112 is misleading. The first several sentences in section 4.1 should be moved here to provide readers an accurate description of the status of science, and the authors need to further elaborate to explain why this study adds values to existing literature.

Response: This has been addressed, see response to previous comment. The revised statements are as : "Recent studies have used a regional climate model to investigate the impact of climate-vegetation interaction for West Africa, identifying significant vegetation feedback in modulating local hydrological cycling (e.g. Alo and Wang, 2010;Wang and Alo, 2012;Yu et al., 2015). Additionally, a number of GCM-based studies have investigated the climate effects of anthropogenic perturbations, such as deforestation or afforestation (e.g. Lawrence and Vandecar, 2015). Such studies point to potentially significant forcing of regional climate dynamics, particularly rainfall patterns, as a result of changes in land cover. No study to date has, however, characterised the coupled dynamics of vegetation and climate under future radiative forcing for the entire African domain at a grid resolution high enough to capture regional features and forcings."

(4) Fig.1: The color scale is very difficult to read if one were to try to figure out the actual magnitude of the model biases. Should use more distinguishable color scales/ use stronger contrast between the colors.

Response: This has been addressed, thanks for the suggestion. Fig 1 is updated accordingly.

(5) Fig.1 and 2 showed severe bias of the model in capturing the spatial pattern of precipitation distribution and vegetation distribution. Essentially, LAI has negligible difference between the Sahelian savannan and the central Africa forest. The discussion and statement about model performance in Section 3.1 significantly downplayed the severity of this model biases.

Response: We agree that these biases are significant. First of all, we have corrected a mistake in data conversion for our post-processing for CRU dataset, in response to comments #4 of reviewer 1. Fig 1&2 are now updated accordingly. We found that model bias as well as observation uncertainty have reduced.

We have added a substantial discussion of the bias in precipitation pattern and LAI in Section 3.1, including acknowledgement for the LAI bias in lines 278-281 "A systematic overestimation is apparent for savannahs, and a significant underestimation for the central Africa rainforest area. These biases in LAI predominantly reflect the corresponding biases in precipitation (**Error! Reference source not found.** b1-b3 and 2c1-c3)" and more explanation for the precipitation bias in lines 247 – 253 "In RCA, the dry bias for annual mean precipitation over central Africa may be partly due to the underestimated daily precipitation during the late afternoon and night in addition to observational uncertainties (Nikulin et al., 2012). The wet bias over the northern savannah is mainly caused by a too early onset of the rainy season (b1, Fig. 2), which is possibly caused by the interactions between the simulated deep convection and the Africa Easterly Waves (Sylla et al., 2011)"

Additionally, in response to Comment (7) from Reviewer #1 which is also relevant to this comments, we have added a new figure (Fig A1) and related discussion evaluating low-level circulation and humidity. We found that the dry bias over central Africa and wet bias over Sahelian savannah are not primarily related to the biases in the circulation, but are more likely to be related to problems with the convection scheme in the regional model (Nikulin et al. 2012). In contrast, the model has done a relatively good job in reproducing overall circulation patterns, including the southern and northern trade wind over Atlantic oceans (Fig. A1), Walker circulation (Fig. 6), which are important for this study. This has been included in section 3.1 (lines 263 - 277).

(6) The model biases in precipitation and more importantly in vegetation could significantly influence the location and magnitude of the difference between FB and NFB, and need to be discussed explicitly.

Response: For the vegetation dynamics, the bias in simulated present-day vegetation is largely related to a bias in precipitation. We agree that this potentially influences the simulated

difference between FB and NFB, as the latter uses the resultant, biased vegetation as forcing. This may lead to an offset in the locations of the strongest impact of vegetation feedbacks in the model, but we assume that this bias does not critically affect qualitative aspects of the feedbacks that we find. One reason for such confidence is that the bias is small in magnitude compared to the size of the simulated future changes in LAI and precipitation. We have add more discussion in section 4.2.

The influence of the bias on model's dynamics is explained in point (8) further below.

(7) Lines 315-320: The albedo difference is negligible? One would think that albedo changes can be significant in areas with increase of vegetation cover.

Response: We agree that albedo changes play a role for surface temperature changes, this was ill-phrased in the original manuscript. We have identified warming effects from albedo change, which gives an overall warming effect in northern hemisphere winter on the edge of the area of forest expansion in the northern savannah region (Fig. A2). An increase in vegetation (forest) cover gives both an albedo (warming) effect and an evaporative (cooling) effect, with the combined effect depending on their seasonal balance. In general, modelling studies tend to show that evaporative cooling effects are more dominant in the tropics while albedo warming effects are more dominant over high-latitude regions (e.g. Claussen et al., 2001;Bala et al., 2007).

The missing explanation for the albedo effect has been added to section 3.3 as "Overall, the turbulent heat fluxes increase, which tends to cool the surface and the lower atmosphere, exceeding the opposing (warming) effects of increased vegetation cover on albedo, thus resulting in an overall cooling effect. Similar behaviour was seen in southern Europe in a previous study with RCA-GUESS (Wramneby et al., 2010).".

(8) Lines 448-453: This is not true. The state of the vegetation is very important in determining the interannual variability of vegetation and the vegetation feedback effects. This is why the issue of severe model bias needs to be acknowledged and its implication explicitly discussed, as suggested in comment 6).

Response: We agree with the reviewer on this point, we did not express our point well in the original manuscript. We intended to point out that bias in LAI within a given land cover type (forest, savannah or grassland) is likely to have a smaller impact on the simulated climate than an inaccurate distribution of land cover types. Although our simulations have evident bias in LAI and precipitation, overall patterns of vegetation distribution across Africa are comparable to observations.

This has been further explained in section 4.3 (lines 512-523) as "Despite biases in the initial precipitation and vegetation state (LAI) for some regions, our model was able to reproduce the present-day land cover type, and the simulated present-date climate is close to previous study (Nikulin et al., 2012) using the same physical sub-model with observed land cover type. Under future climate change, vegetation-induced changes in circulation, thus a substantial change in

moisture transport and precipitation, are mainly triggered by changes in land cover type (Fig. 4a), therefore, we argue that the influences from biases in initial conditions on such mechanism found in this study should be limited. Our study used prescribed SST forcing from a GCM and could thus not account for additional or opposing feedbacks mediated by ocean dynamics. However, as the ocean heat capacity is relatively large and variation in land-ocean thermal contrast can be greatly buffered by ocean heat uptake (Lambert and Chiang, 2007), we suggest that results should not change fundamentally if a dynamic ocean component was introduced to the model."

Minor comments: Lines 92-94: "... are important to … " is rather awkward. You mean "… are important determining factors for …" ? Fig. A4: "temperature gradient" should be changed to "temperature contrast" as y-label.

Response: changed as suggested, thank you for the suggestions.

[revised manuscript text omitted]

---

## Author Response (AR2)

Dear Prof. Dr. Claussen, dear reviewers,

Thank you very much for your assessment and the constructive comments to improve our manuscript. Our response is detailed below, as well as being reflected in the revised manuscript. We hope that we have addressed all concerns and questions that were raised.

With best regards,

Minchao Wu (on behalf of all authors)

**Comments from reviewer #1:**

(1) Re: Response to (1)

In the Rowell (2013) paper, I did not find an evaluation of CanESM2 SSTS – this paper is focused only on the ability of the coupled GCMs to capture the relationship between SSTs and Sahel rainfall (JJAS). According to Fig. 5 in that paper, CanESM2 has particularly weak teleconnections between Sahel rainfall and SSTs, with the exception of the Mediterranean, but I did not find an evaluation of the SST simulation. The Xu paper is only on the southeast tropical Atlantic, and they do not call out the Can ESM2 as being particularly accurate. I really recommend that the authors address this important issue, and consider the possible role of SST biases in precipitation biases.

Response: Thanks for suggesting this. We have added an additional figure (Fig. A1 in the revised manuscript), comparing CanESM2 SST and observation-based dataset (HadISST1.1). Given the imperfect simulated SSTs by CanESM2, we agree that its influences on our study should be considered. We revised the discussion from line 263 as:

"The SST forcing is important for African climate. SSTs used in the study domain present warm biases up to 5°C for the southeast and equatorial tropical Atlantic, and cold biases for the northern and southern sub-tropic Atlantic and the Mediterranean sea (3rd row, Fig. A1). Such warm biases in the tropical Atlantic, especially in JJA, may partly contribute to the overestimated rainfall in Guinea Coast (Fig. 1b3), and the cold biases in sub-tropics may link to the underestimated rainfall in the Sahel and southern African when comparing to the ERA-Interim driven simulation (NFB-RP, Fig. 1b3; 2nd row, Fig. A1). The analyses by Rowell (2013) for the global tropics and by LaRow et al. (2014) and Xu et al. (2014) for tropical sub-regions suggest that the SST biases in CanESM2 are comparable to other CMIP5 models".

(2)Re Responses to (2), (3) and (4)

Thank you for addressing these issues more fully.

Response: You are welcome.

(3)Re Response to (5)

I appreciate the rewording. It doesn't help too much in making people understand that the so-called

"Charney hypothesis" is incorrect, but it's at least more accurate.

Response: Thanks.

(4)Response to (6)

I didn't understand your response – especially the second sentence was not clear. Are you saying the

20 c. vegetation forcing is too small to have an effect? Or the difference in averaging periods is important? What about the 21 st c. conditions will make vegetation forcing more prominent?

Response: We wanted to state that the contrasting influences from the static (RCA) and dynamic (RCA-GUESS) vegetation is very limited in the 20th century. When climate starts to change, the impact of dynamic vegetation becomes visible between prescribed (present-day) vegetation and dynamically changing vegetation. Hence, the comparison of present-day precipitation with a previous version of RCA does not show any evaluation of the importance of vegetation dynamics, but only confirms the new version of RCA to result in similar precipitation.

(5)I stand corrected on the savanna/savannah spelling!

Response:  Thanks.

**Comments from reviewer #2:**

The authors have addressed most of my previous comments.

I do have two additional comments that are minor but important in improving the quality of the paper:

1) This is of course not the first study of this type conducted for this region. It is important that results be compared with those of previous studies, at least qualitatively. Any similarity? Any major differences? This discussion (preferably in the last section of the paper) is important for guiding future studies.

Response: Thanks for suggesting this, some statements to compare previous similar study have been added in the discussion section 4.3, as:

"Similar to the local effects over semiarid region identified in a previous vegetation-feedback study focusing on an African sub-domain (Yu et al., 2015), in which vegetation feedback increases local precipitation, our study in advance presents a remote effect induced by vegetation feedback over the African wet tropics that may not be easy to capture in a smaller simulation domain. A larger domain including sufficient adjacent ocean could enhance land-ocean interaction through changes in regional circulation and allow regional teleconnection features, e.g. the influences of the tropical Atlantic on rainfall in Guinea Coast and central Africa (Camberlin et al., 2001;Nicholson and Grist, 2003), and link between the Mediterranean Sea and Sahelian rainfall (Rayner et al., 2003), to develop"

2) In addressing one of my previous comments about severe model biases, the authors added a paragraph where the biases in the model was referred to "initial state". This is extremely misleading. It leaves the wrong impression that the bias might be just within the model initial conditions, which is not true. The biases shown are biases related to model structure that will stay with the model throughout the whole simulation. Please correct this misuse of terminology.

Response: Thanks for spotting this out. Some words in the last paragraph in section 4.2 are misleading indeed. We have corrected the statement as:

[revised manuscript text omitted]